# Timescales of water accumulation in magmas and implications for short warning times of explosive eruptions

M. Petrelli [1], K. El Omari[2], L. Spina[1], Y. Le Guer[2], G. La Spina [3] & D. Perugini[1]

Water plays a key role in magma genesis, differentiation, ascent and, finally, eruption. Despite the recognized crucial function of water, there are still several issues that continue to blur our view about its role in magmatic systems. What are the timescales of $H_2O$ accumulation in crystallizing magmas? What are the ascent rates of water-rich residual melts leading to explosive eruptions? Here, we track the timescale of water accumulation in a residual melt resulting from crystallization of a hydrous $CO_2$-bearing magmatic mass stored at mid- to deep-crustal levels in a subduction-related geodynamic setting. Our results indicate that, after a repose period ranging from few to several thousand years, water-rich melts with water concentrations larger than 6–9 wt.% can migrate towards the Earth surface in very short timescales, on the order of days or even hours, possibly triggering explosive eruptions with short warning times and devoid of long-term geophysical precursors.

[1] Department of Physics and Geology, University of Perugia, Piazza dell'Università 1, 06123 Perugia, Italy. [2] Université Pau & Pays Adour, Laboratoire des Sciences de l'Ingénieur Appliquées à la Mécanique et au Génie Electrique (SIAME)—Fédération IPRA, EA4581, 64000 Pau, France. [3] School of Earth and Environmental Sciences, The University of Manchester, Manchester, M13 9PL, UK. Correspondence and requests for materials should be addressed to M.P. (email: maurizio.petrelli@unipg.it)

The presence of volatiles in subduction zones plays a fundamental role in many processes involved in magma genesis and differentiation[1–6]. Among them, water is the most abundant, and it has a significant effect on the rheological behavior of magmas[7], on the chemistry of growing crystals[8] and, as a consequence, on the lifetime evolution of a magmatic system[9]. The second volatile species in magmas is carbon dioxide, which also plays a significant role in subduction zones, modulating, for example, the solubility of water[3] and controlling the initial stages of degassing of ascending magmas[4].

Also, at depths where the magmatic pressure is below the saturation threshold, volatile species can exsolve rapidly, potentially causing magma chamber rejuvenation and triggering explosive eruptions[10–16]. The vigor of volatile exsolution can be strongly affected by the ascent rate of volatile-rich magmas, finally exerting a significant role in modulating the eruptive style[6,17,18].

Melt inclusions in minerals can provide useful information about volatile contents in magmatic systems[19,20]. During their growth in magmas, minerals occasionally trap small parcels of melt (preserved as inclusions), providing a record of the volatile content of the system[11,21,22]. Available data from arc volcano mafic magmas point to a rather constant global average of 3.9 ± 0.4 wt.% $H_2O$[22]. This is considered as an indication that the average water content retrieved from melt inclusions reflects the vapor saturation conditions of the last magma storage depth at ~6 km, a characteristic storage depth beneath arc volcanoes[22,23]. Similar to water, carbon dioxide content in melt inclusions is also affected by processes occurring at shallow crustal levels, often resulting in underestimates of the $CO_2$ budget[24]. As a consequence, melt inclusions may not correctly archive information about the $H_2O$–$CO_2$ systematics at deeper levels, hindering de facto a reliable estimation of water and carbon dioxide contents in magmas evolving at middle and lower crustal depths. Recent investigations on nominally dry minerals, such as pyroxene from arc volcanoes, point to the presence of higher $H_2O$ contents than those recorded by melt inclusions[25]. For example, Edmonds et al.[25] hypothesized the existence of composite bodies of water-rich (6–9 wt.%) eruptible magma extending vertically (down to 16 km) within the upper crust beneath the volcanic island of Montserrat (Caribbean Sea). The presence of water-rich magmas ($H_2O$ ≥ 8.0 wt.%) at mid-crustal levels is also supported by recent studies combining geophysical investigations, laboratory experiments, and petrological data to constrain deep electrical conductivity anomalies beneath the Uturuncu Volcano (Central Andes)[26]. Also, similar features have been observed beneath the Cascades volcanic arc and Taupo Volcanic Zone, suggesting that the presence of water-rich magmas could be a common feature of active continental arcs[26]. As for carbon dioxide, Blundy et al.[24] suggest, in agreement with Wallace[5], an average content of ~0.3 wt.%. in arc parental magmas.

Despite the developments mentioned above, our knowledge about the fate of volatile-rich magmas in the Earth crust is still actively debated[2,6] and deserves further investigation. Key questions include: What are the timescales for volatiles accumulation in crystallizing magmas at mid- to deep-crustal levels? What are the ascent velocities of volatile-rich residual melts (produced by the crystallization of volatile-bearing parental magmas at mid- to deep-crustal levels) toward the Earth surface, eventually culminating in explosive volcanic eruptions?

To answer the first question, we present new numerical simulations, targeted at reproducing the thermal and petrological evolution of magmas at mid- to deep-crustal levels. In particular, the thermochemical behavior of a subduction-related, volatile-bearing, magmatic body is investigated.

The model accounts for different initial chemical compositions of the parental magma and initial water contents ranging from 2 to 4.5 wt.% (please refer to the "Methods" for a complete description of the model development and parametrization) both in presence and absence of $CO_2$. It also accounts for different storage conditions (0.7–1.0 GPa corresponding to ~20–35 km in depth) of both closed and open magmatic systems.

In particular, we first investigated a closed magmatic system exchanging heat with the hosting rocks only by conduction. For comparison, an open system consisting of progressive refilling by new parental magma is also modeled. Finally, we studied the effect of the development of convective motions within the system, accounting for the non-Newtonian rheology of the magma during the crystallization[27] and considering Rayleigh numbers (Ra) up to $10^9$. The numerical simulations started at the liquidus temperature (i.e., no crystals are present within the system) and continued until the attainment of a rheological behavior close to the jamming conditions[28], i.e., when the melt extraction from the magmatic mush becomes most probable[29]. The fate of the volatile-rich residual melts segregated from the crystal-rich mushes is also investigated. In particular, we provide a comprehensive discussion of the factors promoting and contrasting the rising of volatile-rich melts in the crust. Finally, the timescales for the transfer of volatile-rich melts toward the Earth surface, eventually culminating in volcanic eruptions, are estimated. Results indicate that water-rich melts can ascend rapidly to the surface over timescales of hours to days with very short warning times.

## Results

**$H_2O$ and $CO_2$ solubilities in arch magmas**. In order to investigate the timescales of volatile accumulation in subduction zones at mid- to deep-crustal levels, the $H_2O$ and $CO_2$ solubilities in mafic primitive arc magmas above the liquidus temperature (i.e., before the beginning of the crystallization process) must be considered. We focus on pressures between 0.7 and 1.0 GPa, corresponding to crustal depths ranging between ~20 and 35 km. Figure 1a displays the saturation conditions for one[30] of the starting magmatic compositions considered here as a proxy for primitive mafic arc magmas[30–33]. The same curves for the other parental magma compositions that we considered are reported as Supplementary Material (Supplementary Fig. 1a, b). Figure 1a shows that, at 1.0 GPa (i.e., ~30–35 km of depth), the melt phase can host ~11 wt.% of $H_2O$ in a $CO_2$-free system. Figure 1a also shows that the maximum solubility of carbon dioxide in the system at 1.0 GPa is ~0.90 wt.%, corresponding to a water content of ~3.8 wt.%. $H_2O$ and $CO_2$ solubilities progressively drop down as pressure decreases (Fig. 1a). In a magmatic system subjected to isobaric cooling (i.e., crystallizing at a constant depth), melt solubilities for water and carbon dioxide will progressively evolve in response to thermal and chemical changes. As an example, Fig. 1b reports the evolution of the $H_2O$ and $CO_2$ saturation conditions for the residual melt of a magmatic system crystallizing at 0.7 GPa. Figure 1b highlights that, during the crystallization process, the maximum solubility of water progressively increases from ~10.2 to ~12.0 wt.%. On the contrary, carbon dioxide solubility gradually decreases from ~0.53 to ~0.36 wt.%.

**Thermodynamic modeling**. Thermodynamic modeling can be also used to track the evolution of $H_2O$ and $CO_2$ during the crystallization of a parental magma at pressures ranging from 0.7 to 1.0 GPa (see the "Methods" section for further details). As an example, we report the cases for magmas characterized by an

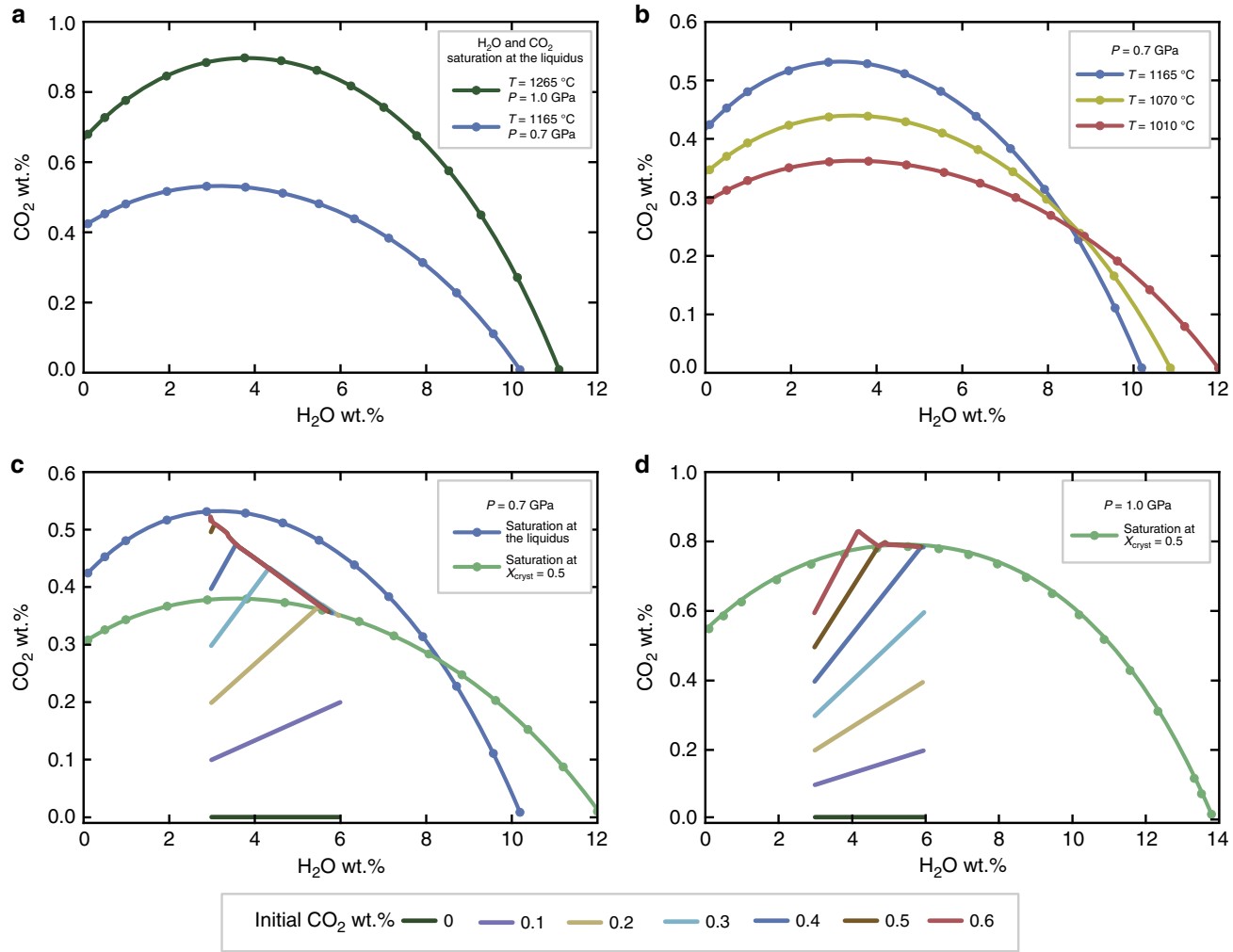

**Fig. 1** Water and carbon dioxide solubilities in the melt phase. Solubility curves for water and carbon dioxide calculated for the mid-MgO system[30]: **a** Solubilities of the parental melt at pressures of 0.7 and 1.0 GPa and liquidus temperatures; **b** Solubilities of the residual melt during the crystallization process at a pressure of 0.7 GPa. The $SiO_2$ content in the residual melt on dry basis is 50.3, 50.9, and 57.2 wt.% respectively; **c** Equilibrium thermodynamical models performed by rhyolite-MELTS[59] at 0.7 GPa displaying the evolution of $H_2O$ and $CO_2$ in the residual melt during the crystallization process; **d** Equilibrium thermodynamical models performed by rhyolite-MELTS[59] at 1.0 GPa displaying the evolution of $H_2O$ and $CO_2$ in the residual melt during the crystallization process. The parameter $X_{cryst}$ reported in **c**, **d** is the crystal mass fraction

initial water content of 3 wt.% and $CO_2$ concentrations varying from 0 to 0.6 wt.% at a pressure of 0.7 (Fig. 1c) and 1.0 (Fig. 1d) GPa, respectively. The melt phase progressively accumulates $H_2O$ and $CO_2$ up to the attainment of the saturation conditions due to the development of $H_2O$- and $CO_2$-free phases in the early evolution of the crystallization process. Hence, the melt starts exsolving a fluid phase, mostly composed of the least soluble volatile component (i.e., $CO_2$). As a consequence, after reaching the saturation conditions, $CO_2$ concentrations are buffered at the saturation levels (~0.33 wt.% in Fig. 1c). In contrast, the evolution of $H_2O$ is slightly affected by the attainment of the saturation conditions. At 1.0 GPa (Fig. 1d), the overall behavior of the system during an isobaric cooling is similar to that observed at 0.7 GPa (Fig. 1c). In this case, due to the higher solubilities for both water and carbon dioxide, the maximum $CO_2$ concentration is ~0.79 wt.% at a crystal mass fraction ($X_{cryst}$) approaching 0.5 (Fig. 1d). The behavior of a parental magma richer in $H_2O$ ($H_2O$ = 4.5 wt.%) is similar; the detailed description of the simulations is provided as Supplementary Material (Supplementary Fig. 2). In this case, the maximum final concentration of carbon dioxide in the melt phase is ~0.8 and 0.3 wt.% for pressures of 1.0

(Supplementary Fig. 2a) and 0.7 GPa (Supplementary Fig. 2b), respectively. The impact of the attainment of volatile saturation conditions on the magmatic system can be tracked monitoring the relative increment of $H_2O$ ($\Delta H_2O$; defined as the ratio between the $H_2O$ content and its initial value) in the melt phase plotted against the melt fraction ($X_{melt}$; Fig. 2a). Figure 2a shows that, at 1.0 GPa, the melt phase of the studied systems (3.0 < $H_2O$ < 4.5 wt.% and 0.0 < $CO_2$ < 0.6 wt.%) accumulates water, behaving essentially as an under-saturated system (i.e., no $H_2O$ release). On the contrary, at 0.7 GPa, $CO_2$-rich melts may release a larger amount of water, slightly deviating from the behavior of the under-saturated system (Fig. 2b). Also, the crystallization of hydrous phases (e.g., hornblende) may potentially affect the process of water accumulation in the melt phase during crystallization of water-rich arc magmas. For example, hornblende is a ubiquitous phase in volcanic and plutonic rocks produced in magmatic arc systems[34]. Although, amphibole does not crystallise massively in the experiments considered here[30–33] (please refer to the "Methods" section for further details), its potential effect on the evolution of the system needs to be considered.

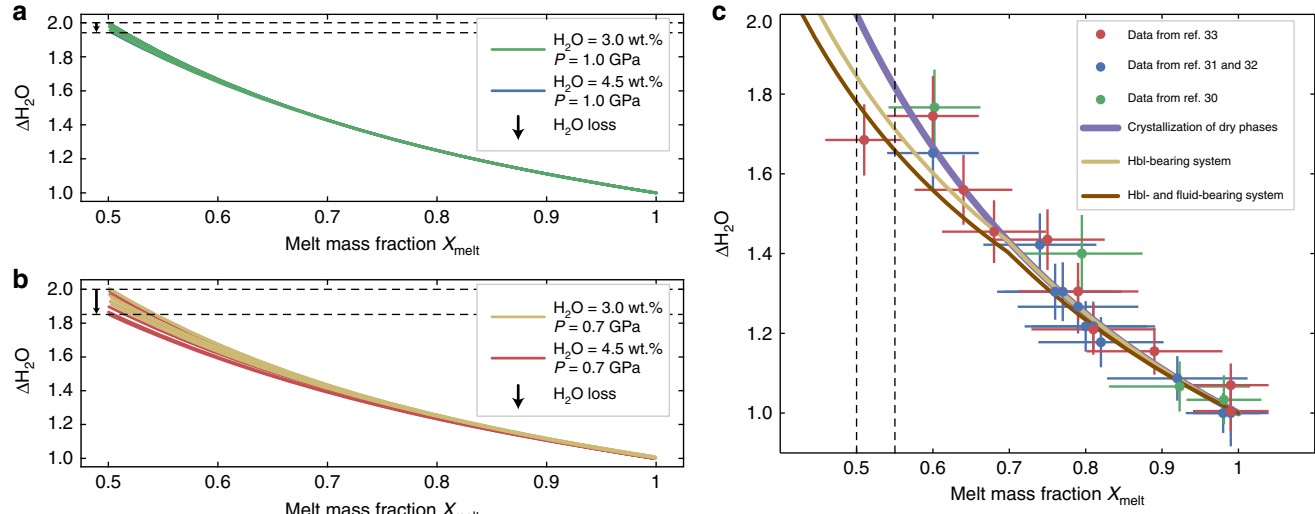

**Fig. 2** Water accumulation in the melt phase during crystallization. Evolution of the $\Delta H_2O$ parameter (calculated as ratio between the water content in the residual melt and the initial $H_2O$ concentration in the system) against the melt mass fraction ($X_{melt}$). **a** Results of equilibrium thermodynamical simulations performed by rhyolite-MELTS[59] showing the evolution of the $\Delta H_2O$ parameter in the residual melt of a system crystallizing at 1.0 GPa. **b** Results of equilibrium thermodynamical simulations performed by rhyolite-MELTS[59] showing the evolution of the $\Delta H_2O$ parameter in the residual melt of a system crystallizing at 0.7 GPa. **c** Mass balance modeling of the $\Delta H_2O$ parameter displaying the evolution of $\Delta H_2O$ parameter during the crystallization of a magmatic system. Different curves are reported for the crystallization of an anhydrous crystal assemblage (purple curve), a hornblende-bearing crystal assemblage (yellow curve), and an over-saturated hornblende-bearing crystal assemblage (brown curve). The data extracted from the investigated petrological data set[30-33] are also reported in the picture with error bars expressed as one standard deviation

**Mass balance modeling**. The evolution of water content in the melt phase during amphibole crystallization can be tracked by mass balance modeling (Fig. 2c; please refer to the "Methods" section for further details). Figure 2c compares the evolution of $\Delta H_2O$ vs. the melt mass fraction ($X_{melt}$) during the crystallization of only dry phases (e.g., the sequence olivine, pyroxene, spinel, and plagioclase), the crystallization of a hornblende-bearing magma (e.g., the sequence olivine, pyroxene, spinel, plagioclase, and hornblende), and a hornblende-bearing over-saturated magma where volatile exsolution also contributes to the process of water subtraction to the melt phase. Data extracted from the used experimental petrology data set[30-33] are also reported in Fig. 2c. The plot shows that crystallization of amphibole does not produce substantial changes in $\Delta H_2O$ of the magmatic system as indicated by the fact that crystallization of amphibole and volatile exsolution might decrease the $\Delta H_2O$ parameter by maximum ~0.2 at the attainment of crystal mass fractions ($X_{cryst}$) of 0.5. This being so, in the following sections we will consider the crystallization of dry phases in an under-saturated system up to the attainment of $\Delta H_2O$ equal to 2.0 (i.e., a crystal mass fraction of 0.5).

**Timescales for $H_2O$ accumulation in the residual melt**. In order to define the timescales for $H_2O$ accumulation, we performed thermal numerical simulations (please refer to the "Methods" section for further details) monitoring the evolution of the $\Delta H_2O$ variable (i.e., the increment of water relative to the initial values) over time. First, we considered the evolution of a closed system, where heat is transported only by conduction (Fig. 3a). In this configuration (Fig. 3a), $\Delta H_2O$ progressively increases with time and attains the value of 2.0 (i.e., $X_{cryst}$ ~0.5) at dimensionless time $\tau$ (defined as $\tau = t\alpha/L^2$) equal to ~0.107. A further set of numerical experiments allowed us to monitor the evolution of a magmatic system periodically refilled by new parental magma (i.e., open

system; Fig. 3b–d). At fast-refilling rates (e.g., a magmatic system that experiences an incremental growth in less than $0.017\tau$; Fig. 3b), the overall behavior is similar to the closed system (Fig. 3a). At these conditions, the threshold value of 2.0 for $\Delta H_2O$ is attained at a $\tau$ equal to ~0.092. At mid-refilling rates (e.g., a magmatic system with a total incremental growth of ~$0.035\tau$; Fig. 3c), $\Delta H_2O$ first shows a dumped oscillation around a value of ~1.6, then starts behaving similarly to the closed system, and it attains a $\Delta H_2O$ value of 2.0 at a $\tau$ equal to ~0.087. At slow-refilling rates (e.g., a magmatic system that completes the incremental growth in ~$0.087\tau$; Fig. 3d), the $\Delta H_2O$ experiences a dumped oscillation around ~1.9, until it stabilizes to a value larger than 2.0 at a $\tau$ equal to ~0.100. The oscillations of the $\Delta H_2O$ parameter are mainly related to the periodic arrival of new magma characterized by a lower $H_2O$ content. Transformation from dimensionless to dimensional units allows us to better understand these timescales. Consider a magmatic system characterized by a thermal diffusion of $\alpha = 8 \times 10^{-7}$ m$^2$ s$^{-1}$ (Table 1) and a lateral dimension ($L$) equal to ~5 km. For the closed system scenario reported in Fig. 3a, the attainment of $\Delta H_2O$ ~2.0 occurs after ~106 ky. Under the same boundary conditions, the open system scenarios (Fig. 3b–d) are characterized by refilling rates of $3.9 \times 10^{-3}$, $1.95 \times 10^{-3}$, and $7.8 \times 10^{-4}$ km$^3$ y$^{-1}$, respectively. These values are in agreement with typical growth rates of magmatic bodies in arc settings[35-37]. For the cases reported in Fig. 3b–d, a $\Delta H_2O$ value of 2.0 is reached after 91, 86, and 100 ky, respectively.

The cases reported in Fig. 3a–d are all representative of magmatic bodies in static conditions (i.e., characterized by physical conditions that inhibit the development of convective motions within the system). Additional experiments will be now presented to evaluate the effects of the development of natural convection within the system.

Figure 3e reports the evolution of $\Delta H_2O$ vs. $\tau$ for a closed magmatic system characterized by Ra numbers equal to $10^7$

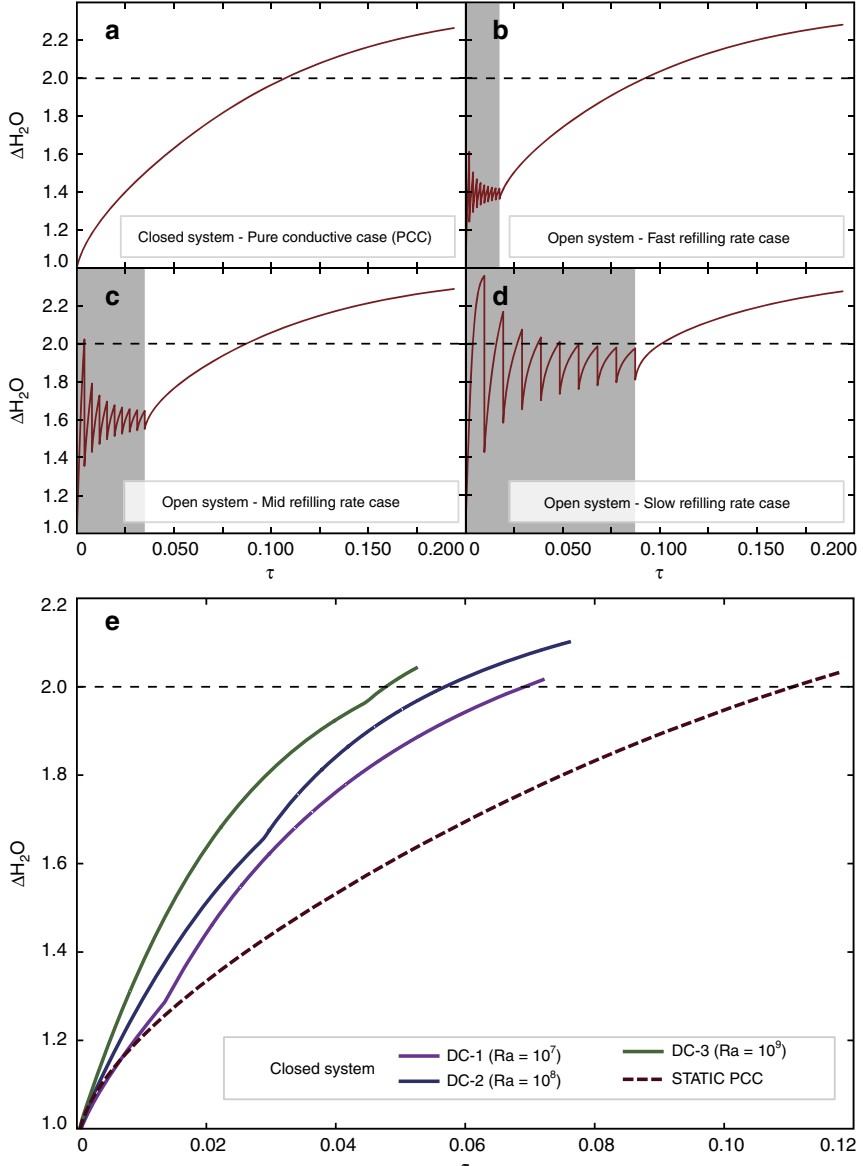

**Fig. 3** Temporal evolution of water concentration in the melt phase. Evolution of the $\Delta H_2O$ parameter (i.e., the ratio between the current $H_2O$ concentration in the residual melt and its initial value) over the non-dimensional time $\tau$, calculated as $\tau = t\alpha/L^2$, for: **a** closed system in static conditions (i.e., convective dynamics do not develop within the system), **b** open system periodically refilled by new magma at fast-refilling rates; **c** open system periodically refilled by new magma at mid-refilling rates; **d** open system periodically refilled by new magma at slow-refilling rates; **e** closed system characterized by the development of buoyancy-driven convective motions. Three dynamical cases are reported for Rayleigh numbers equal to $10^7$ (DC-1), $10^8$ (DC-2), and $10^9$ (DC-3). The pure conductive case (PCC) of **a** is also reported as reference

(dynamic case 1, DC-1), $10^8$ (dynamic case 2, DC-2), and $10^9$ (dynamic case 3, DC-3). The static pure conductive case (PCC) scenario is also reported as a reference. The dynamic cases are dramatically more efficient in accumulating $H_2O$ in the residual melt (Fig. 3e) than the static configurations because of the faster cooling due to convection, resulting in higher crystallization rates for olivine, clinopyroxene, and plagioclase. Residual melts deriving from DC-1, DC-2, and DC-3 reach the $H_2O$ concentration threshold at $\tau$ of ~0.069, 0.057, and 0.048, respectively, up to ~2.3 times faster than the static PCC scenario.

The relationships emerging from the numerical experiments equip us with a new tool to understand the timescales of water accumulation in subduction-related residual melts. In dimensional terms, the following relationships, linking the size of the magmatic reservoir ($L$) and the time ($t$) required to reach the

threshold water content (i.e., $\Delta H_2O$ larger than 2.0), hold:

$$PCC : \log_{10}[t] = 2\log_{10}[L] + 0.63, \tag{1}$$

$$DC-1 : \log_{10}[t] = 2\log_{10}[L] + 0.44, \tag{2}$$

$$DC-2 : \log_{10}[t] = 2\log_{10}[L] + 0.35, \tag{3}$$

$$DC-3 : \log_{10}[t] = 2\log_{10}[L] + 0.28, \tag{4}$$

where $L$ and $t$ are expressed in km and ky, respectively. Concerning a possible natural scenario, these equations indicate that a magma body with a lateral dimension ($L$) equal to 5.0 km will attain the water concentration threshold in 69, 56 and 48 ky

| Table 1 Initial thermophysical properties of the magmatic system | | |
|---|---|---|
| **Property** | **Value range** | **Unit** |
| Density, $\rho_r$ | 2540–2750 | kg m$^{-3}$ |
| Dynamic viscosity, $\eta_r$ | 1.2–8 | Pa s |
| Thermal diffusivity, $\alpha_r$ | $8.0 \times 10^{-7}$–$8.7 \times 10^{-7}$ | m$^2$ s$^{-1}$ |
| Thermal conductivity, $k_r$ | 2.2 | W m$^{-1}$ K$^{-1}$ |
| Specific heat, $c_{p,r}$ | 1000 | J kg$^{-1}$ K$^{-1}$ |
| Latent heat of crystallization, $L_m$ | $3.5 \times 10^5$ | J kg$^{-1}$ |
| Temperature difference, $\Delta T$ | 105–190 | K |
| Prandtl number, Pr | 545–3636 | – |
| Stephan number, Ste | 0.3–0.54 | – |

Ranges of thermophysical properties of the magmatic system at the initial state as reported in Supplementary Table 1

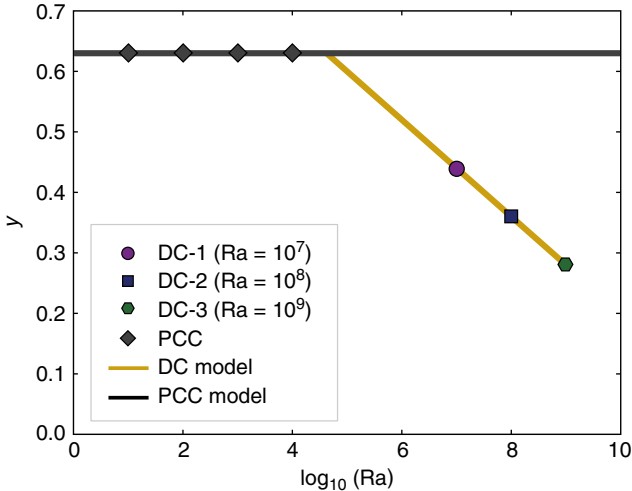

**Fig. 4** Timescales of water accumulation in the melt phase. Relation between intercepts of Eqs. (1), (2), (3), and (4) and $\log_{10}[\text{Ra}]$. Data fitting is performed with the piecewise function (5)

for the DC-1, DC-2, and DC-3, respectively, which are considerably faster than the PCC, indicating the fundamental role of magma chamber dynamics in accelerating the production of water-rich melts.

Equations (1), (2), (3), and (4) indicate that the only variable changing with Ra is the intercept of the lines in the log–log space (0.28, 0.35, 0.44, and 0.63 for the DC-1, DC-2, DC-3, and PCC, respectively). The PCC model (Eq. (1)) can be considered as representative for systems with low Ra values (i.e., lower than those needed to trigger convective dynamics sufficiently strong to cause a departure from a purely conductive temperature gradient[38]; Fig. 4).

Results reported in Fig. 4 can be described by the piecewise function:

$$y = \begin{cases} 0.63 & \log_{10}[\text{Ra}] < 4.6 \\ -0.08 \cdot \log_{10}[\text{Ra}] + 1 & \log_{10}[\text{Ra}] \geq 4.6, \end{cases} \quad (5)$$

allowing for a generalization of Eqs. (1), (2), (3), and (4) to a single relation:

$$\log_{10}[t] = 2\log_{10}[L] + y, \quad (6)$$

which is valid for Ra ≲ $10^9$ and obeys the initial and boundary conditions imposed for the modeling (see "Methods" section).

## Discussion

Numerical experiments indicate that the time evolution of $H_2O$ concentration is strongly dependent on the dynamic conditions of the magmatic system, with convection playing a key role.

Figure 5 compares the variation of viscosity as a function of temperature for crystal-bearing magmas with a constant $H_2O$ content of 3.0 wt.% (Fig. 5a) and 4.5 wt.% (Fig. 5b) at 0.7 and 1.0 GPa, respectively. Different aspect ratios ranging from 1 to 3 have been investigated for forming minerals, in agreement with aspect ratios expected during crystallization of olivine, pyroxene, and amphibole[37]. Notably, the rheological behavior of the crystal-bearing magmas does not change significantly for these aspect ratios (Fig. 5). Also, the adopted rheological model[28] agrees well with that reported by Laumonier et al.[37]. The viscosity of the hydrous residual melts (where $H_2O$ is progressively accumulating with time) are also reported in Fig. 5. At the attainment of the jamming conditions (i.e., close to the maximum packing fraction), the viscosities of the residual melts are ~40–50 and 7–10 Pa s, respectively. These values, calculated following Giordano et al.[39], are several orders of magnitude lower than those of the crystal-bearing magma (hydrous melt phase plus crystals)[28]. In agreement with Mader et al.[28], the jamming condition occurs at a

crystal volume fraction ($\varphi$) of ~0.5, which represents the lower boundary of the crystallinity window ($0.5 < \varphi < 0.7$), where the extraction of melt from the crystal mush is statistically most probable[29]. In this scenario, the produced residual melts are characterized by very low viscosities ($\eta$ equal to 7–10 and 40–50 Pa s, respectively) and high water contents, which make them suitable for migrating to shallower crustal levels. This implies that the ascending melt can potentially refill shallow magmatic reservoirs[15,40,41] or rise directly to the Earth surface causing an eruption[42].

To further investigate this issue, melt migration velocities were estimated using two different approaches: the buoyancy-driven[43,44] and magma-driven[45,46] ascending dike theories (Fig. 6; see "Methods" section for further details). In the first case, propagation is driven only by buoyancy forces, while in the latter the excess pressure of a reservoir connected to the dike is also considered. Critical solidification velocity $v_c$ were also computed. Ascending velocities below $v_c$ prevent the ascending magma to reach the surface before its solidification. In contrast, velocities larger than $v_c$ allow the magma to migrate to shallow crustal levels, eventually triggering an eruption[47].

Two main scenarios are portrayed in Fig. 6. The first scenario is a melt ascending from a depth of 25 km (Fig. 6a, c) and characterized by a viscosity of 50 Pa s. The second scenario considers a melt moving from 35 km depth (Fig. 6b, d) with a viscosity of 10 Pa s. These viscosity values are in agreement with the results reported in Fig. 5a. Figure 6a, b indicates that buoyancy-driven, water-rich, magmas produced at mid- to deep-crustal levels, can arrive at shallow crustal levels with velocities ranging from 0.15 to 8 m s$^{-1}$. However, many studies suggest that the buoyancy is unlikely to be the sole process driving magma to the surface[44]. Indeed, in the magma-driven formulation, the dynamic of the crack propagation is governed by both the buoyancy of the fluid and the reservoir's excess pressure[44]. Figure 6a, b indicates that magma-driven, water-rich melts can reach the Earth surface with velocities ranging from ~1.8 to over ~20 m s$^{-1}$ for overpressures ($\Delta P_{excess}$) ranging from 2 to 3 MPa. These overpressures are in agreement with typical $\Delta P_{excess}$ values estimated for natural magmatic systems[44,45]. Enhanced ascent rates are also expected due to magma vesiculation at shallow levels (below ~10 km), where the presence of gas bubbles contribute to lower melt density and, therefore, enhance the buoyancy effect[46]. We estimated the effect of magma vesiculation (i.e., compressible fluid

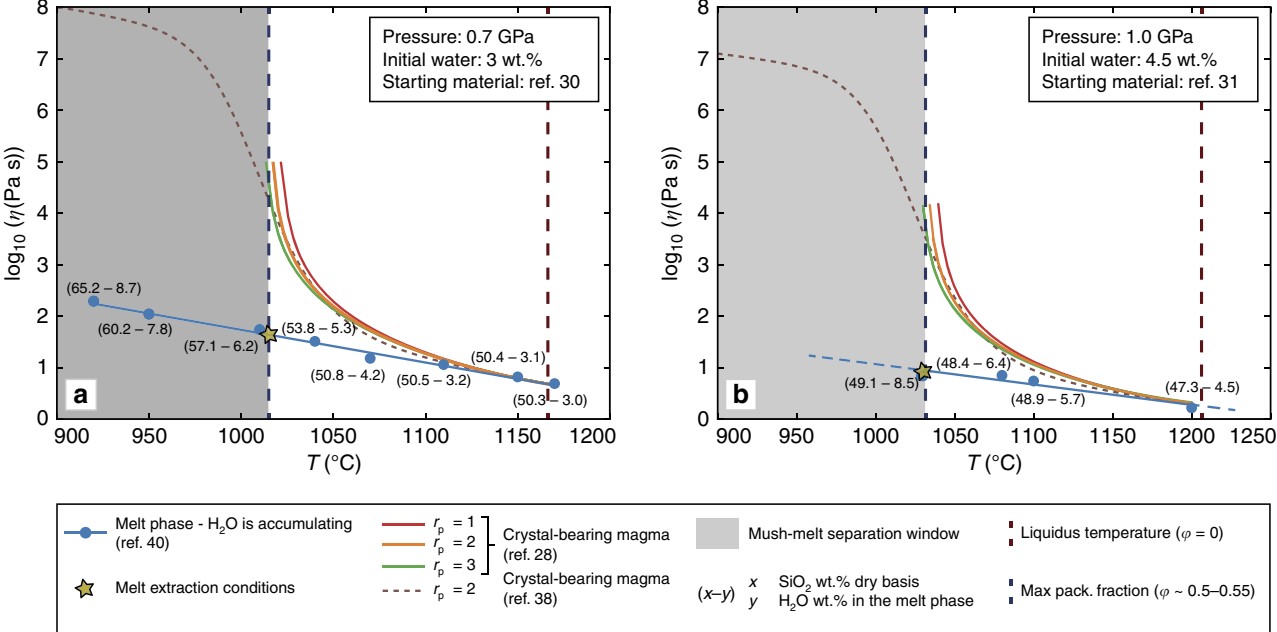

**Fig. 5** Rheological evolution of the magmatic system. Temperature dependency of viscosity for crystal-bearing magma[28,37] and the residual melt[39] at pressures of **a** 0.7 GPa and **b** 1.0 GPa. The crystal-bearing magma has been modeled using the model reported by Mader et al.[28] for different aspect ratios ($r_p$, ranging from 1 to 3, red line for $r_p = 1$, orange line for $r_p = 2$, and green line for $r_p = 3$) of the crystallizing phases. The model reported by Laumonier et al.[37], valid for $\varphi$ values up to ~0.6, is also reported for $r_p = 2$ (brown dashed line). The evolution of the hydrous residual melt with increasing $H_2O$ content due to crystallization has been also reported (calculated in accordance with Giordano et al.[39], not accounting for viscosity variations induced by pressure changes). The gray box marks the area where melt-mush separation is statistically most probable[29]. Finally, the yellow star marks the $\eta$ value of the water-rich melt when the separation between the melt and the crystal-bearing magma is expected (gray area)

behavior) in agreement with Taisne and Jaupart[46] (Fig. 6c, d). Figure 6c, d highlights a significant increase of rising velocities at depths lower than ~10 km. In particular, ascending velocities are increased by a factor ~2 and 10 at depths corresponding to ~4 and 1 km, respectively. The obtained magma ascent velocities are in agreement with estimates (velocity in the range 0.1–25 m s$^{-1}$) provided for natural systems[48–54]. As an example, Lloyd et al.[50] suggested rising velocities up to ~20 m s$^{-1}$ for Volcán de Fuego eruption occurred on October 17, 1974, based on multi-volatile profiles obtained from olivine-hosted melt embayments.

Another process that can affect the ascent velocity, playing the opposite role of buoyancy, is decompression-driven melt crystallization[41]. The increase of the crystal volume fraction during ascent produces an increase of the viscosity of the mixture, which, in turn, generates a decrease in the ascent velocity. The petrological evolution of ascending water-rich melts was considered by Annen et al.[41] who postulated a viscous death of water-rich ascending magmas at depths of ~10–15 km due to depressurization-induced crystallization. The model proposed by Annen et al.[41] successfully describes the system at physical conditions close to the thermodynamical equilibrium. However, the estimated rising velocities and those reported in the literature for natural systems suggest that the ascending magmas could not be at equilibrium conditions[52]. In particular, La Spina et al.[52] indicated that basaltic magmas ascending from ~10 km depth need approximately 2 h to reach the equilibrium crystal volume content. Therefore, when the average ascent velocity is higher than ~1 m s$^{-1}$, the system does not have enough time to reach the equilibrium condition, resulting in a delayed crystallization and leading to a lower viscosity of the mixture. These estimates indicate that, if the ascent velocity is high enough, crystallization can be inhibited and, therefore, water-rich ascending magmas can readily reach the Earth surface.

A potential scenario for the evolution of a crustal arc section is illustrated in Fig. 7. An initial batch of magma containing ~3–4.5 wt.% of water, stored at mid- to deep-crustal levels (~20–35 km), can evolve to a water-rich ($H_2O$ equal to ~6–9 wt.%) residual melt in timescales ranging from a few to several thousand years (see Eq. (6)). These results are in agreement with typical timescales obtained using crystal residence times based on $^{238}U$–$^{230}Th$ and $^{230}Th$–$^{226}Ra$ disequilibrium[40]. If the ascent velocities are too slow (e.g., below the critical velocity), magma storages at mid- to shallow-crustal levels feeding the incremental growth of batholiths might be favored[41,55]. On the contrary, if rising velocities exceed $v_c$ and are large enough to maintain the system far from the thermodynamic equilibrium, water-rich magmas can reach the Earth surface in short timescales (of the order of days or even hours)[42].

In the most hazardous scenario (Fig. 7), after a storage of several thousand years at mid- to deep-crustal levels, water-rich melts can migrate to the surface in a few hours, possibly triggering explosive eruptions with little warning time and devoid of long-term geophysical warnings that can be used for alerting purposes[6,56,57].

Our results suggest a strategy for the risk mitigation of these sudden eruptions. Geophysical investigations should be targeted at detecting the presence of crystal-rich magmatic mushes at mid- to deep-crustal levels characterized by water-rich residual melts, as well as recognizing possible signs of water-rich magmatic systems deep within the Earth crust that might rapidly evolve toward explosive eruptions.

## Methods
**Conceptual model.** The evolution of a magmatic system crystallizing at mid- to deep-crustal levels was modeled considering two main scenarios: a closed system where the magmatic body remains isolated from external magmatic inputs and an

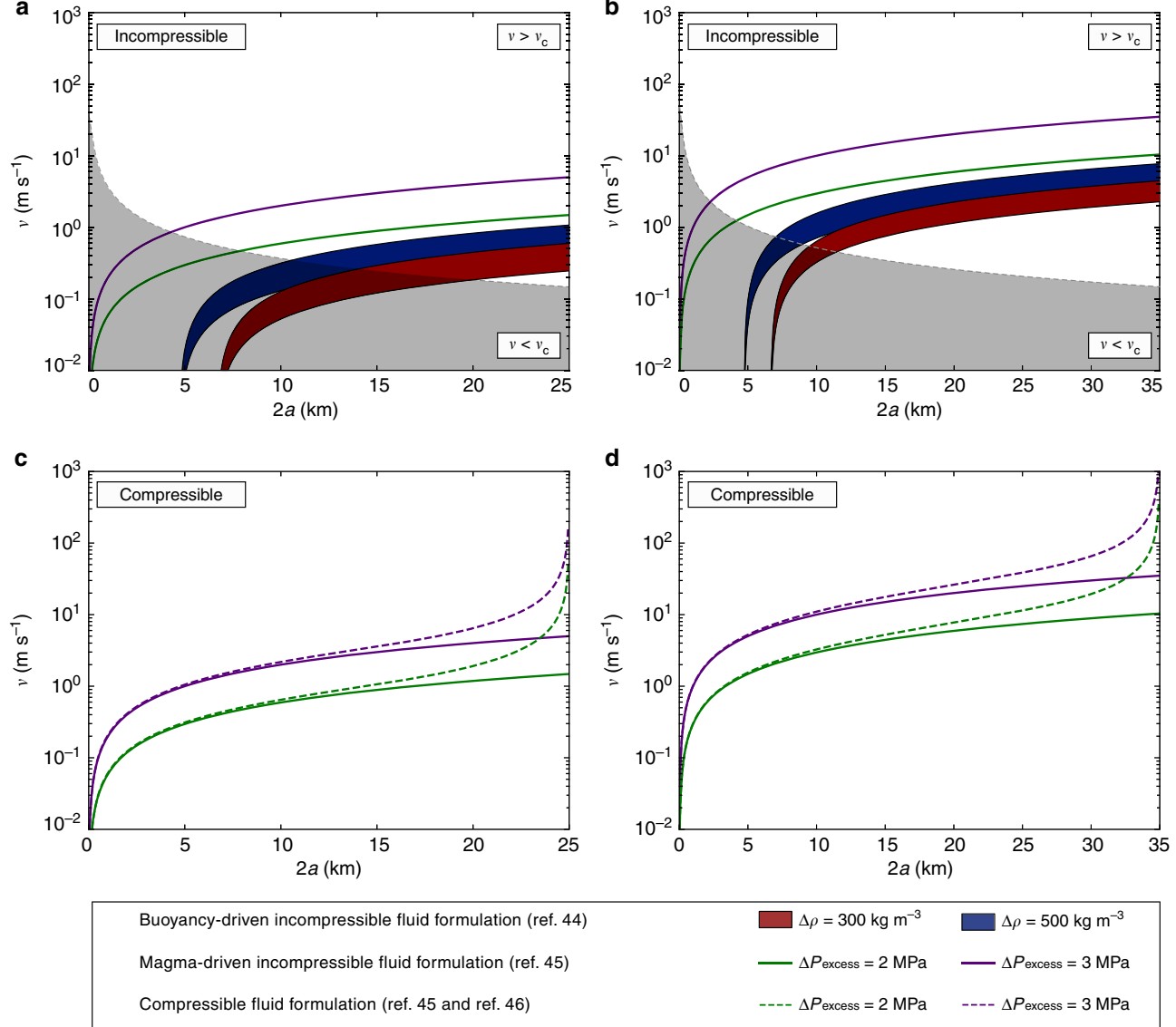

**Fig. 6** Melt migration velocity. Migration velocities of the water-rich melts in a dike rising from 25 km (**a**, **c**) and 35 km (**b**, **d**). The melt is modeled utilizing the buoyancy-driven[43] and the magma-driven[45] formulation and assumed to behave as incompressible fluid in **a**, **b**. The effect of volatile exsolution (compressible fluid) on melt migration velocities has been modeled following Taisne and Jaupart[46] and reported in **c**, **d**

open system where the magma chamber is periodically refilled by new parental magma (Fig. 8).

The initial conditions for the closed system scenario are schematically shown in Fig. 8a. It consists of a two-dimensional (2D) rectangular domain initially filled with a magma at the liquidus temperature[27] (Fig. 8a). During its evolution, the magmatic system cools due to heat exchange with the host rock (Fig. 8c–f). The open-system behavior is simulated in agreement with the progressive incremental growth model proposed by Annen et al.[41,55].

The initial configuration for the open-system scenario (Fig. 8b) consists of a magmatic sill initially emplaced at its liquidus temperature (Fig. 8b) and progressively grows by the periodical addition of new sills at the base of the system (Fig. 8g–j)[41,55].

**Petrological data set**. The petrological data set considered for the parametrization used in the present study consists of published experimental data dealing with crystallization of hydrous magmas at mid- to deep-crustal levels (~20–35 km in depth) corresponding to pressures between 0.7 and 1.0 GPa[30–33]. As the focus is on hydrous magmas, experiments were filtered to consider only those where the initial water content was between 2.0 and 4.5 wt.% (Supplementary Table 1). This allowed us to develop a model parametrization based on a comprehensive set of 26 experiments[30–33] with starting materials having compositions of primitive hydrous mafic arc magmas crystallizing at mid- to deep-crustal levels (Supplementary

Table 2). In detail, the starting materials are a primitive high-MgO (14.6 wt.% MgO) basalt from Soufrière (St. Vincent)[31,32], a near-primary olivine–tholeiite mid-MgO (8.7 wt.% MgO) dike composition from the southern part of the Tertiary Adamello batholith (Northern Italy)[30], and a relatively magnesian basalt (8.7 wt.% MgO) typical of mafic magmas erupted in the Cascades, near Mount Rainier (Washington, USA)[33]. The investigated starting materials represent typical compositions for super- and near-liquidus magmas in subduction-related environments[30]. The experiments are described in detail in literature[30–33] and they are characterized by two key features. The first is that dry phases dominate the crystallization process before attaining the jamming conditions (i.e., a crystal volume fraction, $\varphi$, close to 0.5). The second is that the first hydrous phase is always hornblende[30–33], mostly appearing in experiments at crystal volume fractions larger than 0.5.

**H₂O and CO₂ solubility model**. The solubilities and the saturation conditions for $H_2O$ and $CO_2$ in the melt phase were estimated using the formulation proposed by Ghiorso and Gualda[58]. It consists of a thermodynamic model for estimating the saturation conditions of $H_2O$–$CO_2$ mixed fluids in multi-component silicate liquids[58]. As reported by Ghiorso and Gualda[58], the model is calibrated from published experimental data on water and carbon dioxide solubility, and mixed fluid saturation in silicate liquids[58]. The model assumes that $H_2O$ dissolves to form a hydroxyl melt species, and that $CO_2$ dissolves to compose both a molecular species

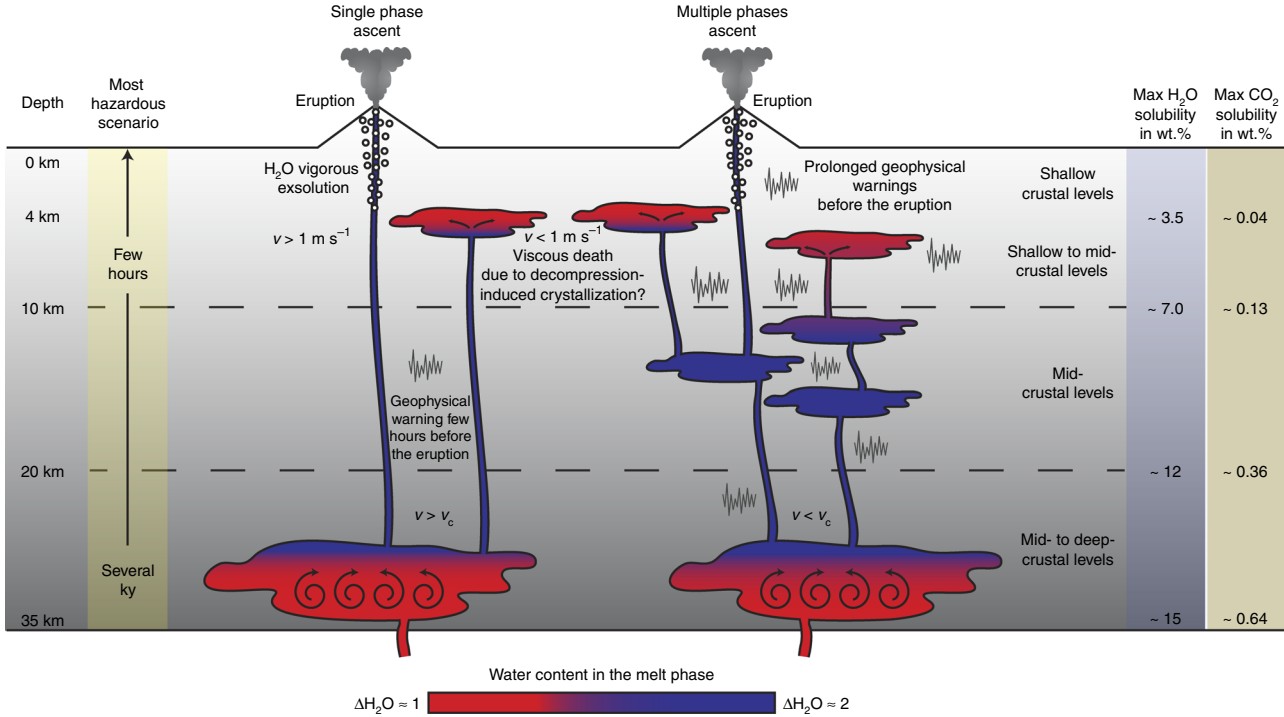

**Fig. 7** Storage timescales and melt rising velocities. Idealized continental crust section reporting the potential storage times at mid- to deep-crustal levels and the estimated timescales for the rising of a hydrous residual melt. Maximum water and carbon dioxide solubilities are calculated on the residual melt at 1010 °C of the mid-MgO experiments[30] in accordance with the model reported by Ghiorso and Gualda[58]. The multiple phases ascent scenario (right portion) is in agreement with literature data (e.g., Annen et al.[41,55]). The single phase ascent scenario is reported on the left portion of the diagram. Here, the most hazardous scenario, with geophysical evidences of volcanic unrest occurring a few hours before the eruption is also shown

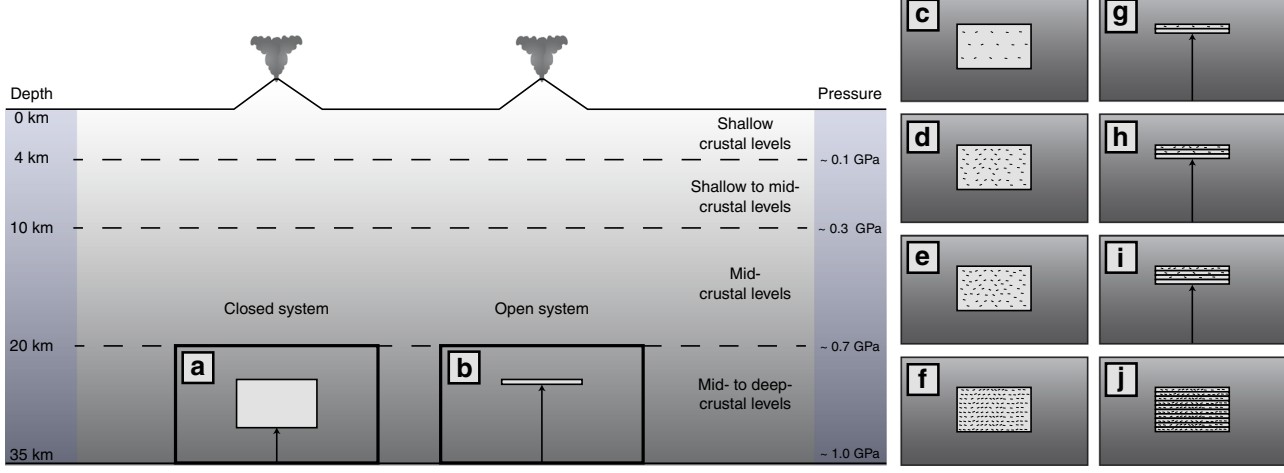

**Fig. 8** Conceptual model for the thermal numerical simulations. **a** the closed-system scenario; **b** the open-system scenario. The evolution of the closed system is schematically described in **c–f**. The evolution of the open system is described in **g–j**

and a carbonate ion, the latter complexed with calcium[58]. The model is restricted to liquids of natural compositions over the pressure range 0–3 GPa making it suitable for the study of magmatic system crystallizing at pressures ranging between 0.7 and 1.0 GPa. It is also suitable to study the $H_2O$ and $CO_2$ solubilities during the rising of magmas from 1.0 GPa to the Earth surface[58].

**Thermodynamical evolution of $H_2O$ and $CO_2$ in the melt phase**. The evolution of $H_2O$ and $CO_2$ in the melt phase during the crystallization process have been modeled using rhyolite-MELTS 1.2[58,59]. Rhyolite-MELTS is a software package that combines the potentials of the MELTS package for thermodynamic modeling with the $H_2O$–$CO_2$ solubility model proposed by Ghiorso and Gualda[58].

Specifically, to model the evolution of $H_2O$ and $CO_2$ during the crystallization process at mid- to deep-crustal levels, we used the dry chemical compositions of the starting materials reported in Supplementary Table 2 and we added different amount of water and carbon dioxide in the ranges of 2 < $H_2O$ < 4.5 wt.% and 0 < $CO_2$ < 0.6 wt.%, respectively. Simulations were started at the liquidus composition and stopped at a crystal mass fraction ($X_{cryst}$) equal to ~0.5. The NNO buffer has been adopted to constrain the oxygen fugacity, and the numerical experiments have been performed at equilibrium conditions fractionating the solid phase.

**Mass balance modeling**. Nominal $H_2O$ concentrations in the residual melts were estimated by the mass balance method, an application of the mass conservation law

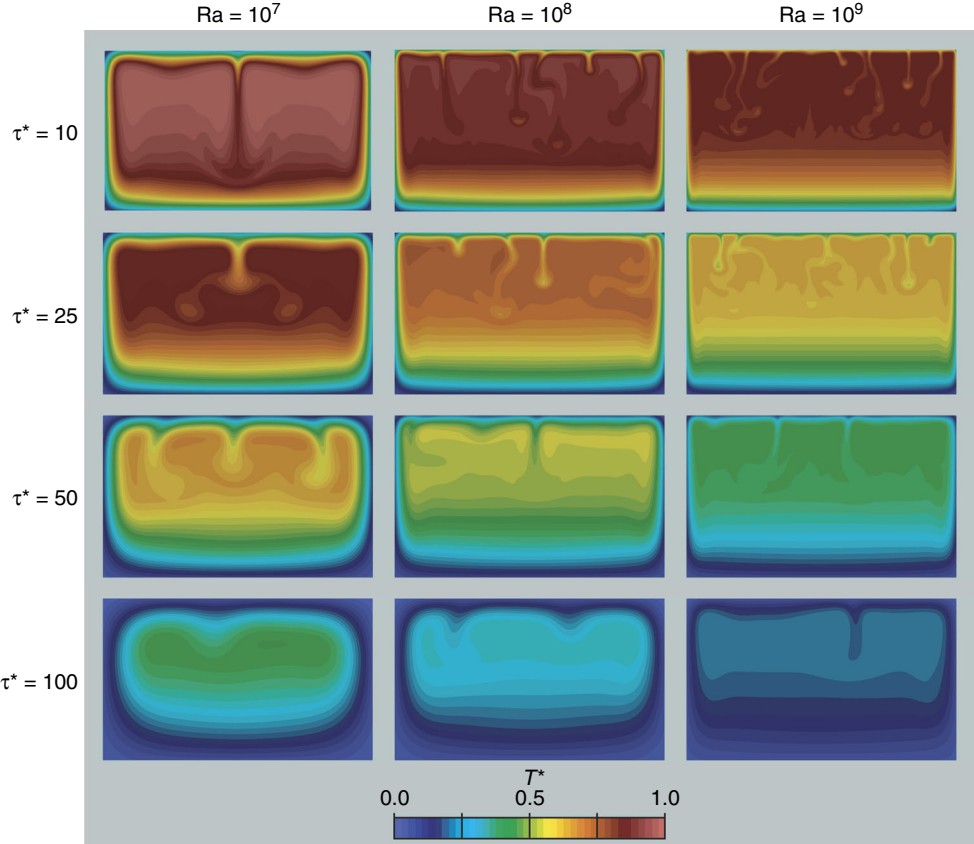

**Fig. 9** Thermal evolution of the magmatic system. Snapshots of temperature field in the magma chamber for Ra = $10^7$, $10^8$, and $10^9$ at different times $\tau^*$. $T^*$ is the dimensionless temperature (see Eq. (15))

that accounts for the material entering and leaving the system[60]. In the specific case of a crystallizing magmatic system, the mass balance equation for $H_2O$ can be expressed as follow:

$$C_{tot}^{H_2O} = \sum_{i=1}^{N} C_i^{H_2O} X_i, \qquad (7)$$

where $C_{tot}^{H_2O}$ is the total $H_2O$ concentration in the system, $N$ is the number of phases able to host a significant amount of $H_2O$, $C_i^{H_2O}$ is the $H_2O$ concentration in the phase $i$ and $X_i$ is its mass fraction. In the studied system (i.e., a magmatic system crystallizing at mid- to deep-crustal levels in subduction-related environment), the only phases able to host a significant amount of $H_2O$ are the residual melt, hornblende (hbl), and the fluid phase[30–33]. Although biotite can potentially host a significant amount of water, it starts crystallizing only at later stages much beyond the attainment of a crystal mass fraction equal to 0.5. As our numerical experiments stop when $X_{cryst}$ reaches 0.5, modeling of this mineral phase is not considered. The mass balance Eq. (7) can be written as:

$$C_{tot}^{H_2O} = C_{melt}^{H_2O} X_{melt} + C_{hbl}^{H_2O} X_{hbl} + C_{fluid}^{H_2O} X_{fluid}, \qquad (8)$$

that yields:

$$C_{melt}^{H_2O} = \frac{C_{tot}^{H_2O} - C_{hbl}^{H_2O} X_{hbl} - C_{fluid}^{H_2O} X_{fluid}}{X_{melt}}. \qquad (9)$$

Finally, if the total concentration of water does not change with time in the system $\left(C_{tot}^{H_2O} = C_{init}^{H_2O}\right)$ the relative $H_2O$ increment to the initial composition ($\Delta H_2O$) is:

$$\Delta H_2O = \frac{C_{melt}^{H_2O}}{C_{init}^{H_2O}} = \frac{C_{init}^{H_2O} - C_{hbl}^{H_2O} X_{hbl} - C_{fluid}^{H_2O} X_{fluid}}{C_{init}^{H_2O} X_{melt}}. \qquad (10)$$

The mass balance has been utilized in the present study to model the evolution of $\Delta H_2O$ during the crystallization process. In detail, in the case of a volatile under-

saturated system crystallizing anhydrous phases only, Eq. (10) reduces to (purple line in Fig. 2):

$$\Delta H_2O = \frac{1}{X_{melt}}. \qquad (11)$$

In order to test the maximum potential impact of hornblende crystallization on the water content of the magmatic system, a model has been developed considering the appearance of this mineral at $X_{melt}$ equal to 0.7 (Fig. 2). The modal proportion of hornblende was assumed to increase linearly from 0 to 30% when $X_{melt}$ passes to 0.4. Note that this is an extreme case and it is in agreement with experiments characterized by high pressure and high water contents ($P = 1.0$ GPa and $H_2O = 4.5$ wt.%) reporting hornblende abundances of 0 and 34% at $X_{melt}$ equal to 0.74 and 0.46, respectively[31,32]. We use this model to test the maximum amount of water that can be potentially subtracted by hornblende in these extreme conditions.

A further test was performed to evaluate, in addition to hornblende crystallization in the framework of the extreme case described above, the loss of $H_2O$ in an over-saturated system at 0.7 GPa considering the results from thermodynamical models. In detail, the results of the numerical simulation reported in Fig. 2b were used to constrain the maximum mass fraction of $H_2O$ potentially subtracted to the system by volatile loss. These values are then introduced in Eq. (10) and reported in Fig. 2c (brown line). The latter must be considered as a very extreme case, as it combines the maximum amount of water potentially subtracted by the development of the hornblende phase at high pressure and high water contents with those related to volatile exsolution at lower pressures (i.e., 0.7 GPa).

**Numerical model for the thermal evolution of the systems.** The numerical model adopted for the study of the thermal evolution of the system in the dynamic case is governed by the conservation equations (mass, momentum, and energy)

reported below in their dimensionless form:

$$\nabla \cdot \mathbf{V} = \mathbf{0}, \tag{12}$$

$$\frac{\partial \mathbf{V}}{\partial \tau^*} + (\mathbf{V} \cdot \nabla)\mathbf{V} = -\nabla P + \nabla \cdot \left( \eta^* \left( (\nabla \mathbf{V}) + (\nabla \mathbf{V})^{\mathbf{t}} \right) \right) + \mathrm{Gr}\, T^* \mathbf{g}, \tag{13}$$

$$\frac{\partial T^*}{\partial \tau^*} + \nabla(T^* \mathbf{V}) = \nabla \cdot \left( \frac{1}{\mathrm{Pr}} \nabla T^* \right) - \frac{1}{\mathrm{Ste}} \left( \frac{\partial f}{\partial \tau^*} + \nabla(f\mathbf{V}) \right), \tag{14}$$

where $\eta^* = \frac{\eta}{\eta_r}$ is the non-dimensional dynamic viscosity; the index r indicates that the considered thermophysical property is taken at the initial reference temperature $T_r$. The system does not consider multi-phase flows, i.e., separate flows of melt and solid phases are not accounted for. The dimensionless temperature is defined as:

$$T^* = \frac{T - T_\infty}{T_r - T_\infty} = \frac{T - T_\infty}{\Delta T_r}, \tag{15}$$

where $T_\infty$ and $T_r$ represent the far-field and the initial temperature, respectively. The volume fraction of the melt phase is given by $f$ ($f = 1 - \varphi$, where $\varphi$ is the crystal volume fraction). The parameter $f$ can be linked to the melt mass fraction ($X_{melt}$) by the following relation:

$$X_{melt} = \frac{f\rho_{melt}}{f\rho_{melt} + \varphi\rho_{cryst}} = \frac{f \cdot R_\rho}{f \cdot R_\rho + 1 - f}, \tag{16}$$

where $\rho_{melt}$ and $\rho_{cryst}$ are the densities of the melt phase and the crystal assemblage, respectively, whereas $R_\rho$ is equal to $\rho_{melt}/\rho_{cryst}$. The Boussinesq approximation is assumed in the above equations, i.e., the temperature-dependence of the density is accounted for in the sole buoyancy term of the momentum equation[61]. The above dimensionless equations have been obtained using a reference velocity $U_r = \nu/L_r$, which is the viscous diffusion velocity, whereas the width of the chamber was taken as the reference length $L_r$. The reference unit time and the dimensionless time are then taken as $t_r = L_r/U_r$ and $\tau^* = t/t_r$, respectively. Equations (12), (13), and (14) show three non-dimensional numbers, the Grashof number (Gr), the Prandtl number (Pr), and the Stephan number (Ste). These numbers characterize the relative importance of buoyancy forces to viscous forces (Gr) in the magma chamber, the ratio of momentum and thermal diffusivities of the magma (Pr) and the ratio of the sensible to the latent heat (Ste), respectively. The product of the Grashof and Prandtl numbers corresponds to the Rayleigh number. The Prandtl number is defined as follows:

$$\mathrm{Pr} = \frac{\eta_r c_{p,r}}{k_r}. \tag{17}$$

The last term in the heat transport equation expresses the release of heat during crystallization of minerals in the magma, which is inversely proportional to the Stephan number and directly proportional to the variation of the liquid fraction $f$ with time ($\partial f / \partial \tau^*$). The Stephan number is:

$$\mathrm{Ste} = \frac{c_{p,r} \Delta T_r}{L_m}, \tag{18}$$

where $L_m$, the latent heat of crystallization for the basaltic magmatic mass. The strength of the buoyancy forces can be characterized by the Grashof number or by the Rayleigh number:

$$\mathrm{Ra} = \mathrm{Gr}\,\mathrm{Pr} = \frac{\rho_r |\mathbf{g}| \beta \Delta T_r L_r^3}{\alpha_r \eta_r}, \tag{19}$$

which is fluid dependent. In Eq. (19), the expansion coefficient $\beta$ takes into account both thermal expansion and crystallinity changes within the system and can be expressed as follow[62]:

$$\beta = \left( \frac{\rho_{cryst} - \rho_{melt}}{\rho_r} \right) \frac{\partial \varphi}{\partial T} + \beta_t, \tag{20}$$

where $\beta_t$ is the thermal expansion coefficient.

The effective temperature difference, $\Delta T$, driving convection for a magmatic system under cooling[63] is linked to the temperature difference between the roof of the magma chamber and the mean temperature of the convecting magma[64]. Thus, the Rayleigh number will decrease rapidly during the cooling of the magma chamber due to the decrease of the temperature difference, as well as to the evolution of the thermophysical properties of the magma (mainly viscosity). Equations (12), (13), and (14) have been solved by direct numerical simulation using the in-house Tamaris CFD code[65]. It is an unsteady finite volume parallel solver for anisothermal incompressible fluids flow based on the SIMPLE algorithm and on second-order accurate spatial and temporal schemes[65]. The

Herschel–Bulkley model was used to describe the magma rheological behavior:

$$\eta = K(T,\varphi)\dot{\gamma}^{n-1} + \frac{\tau_0(\varphi)}{\dot{\gamma}}, \tag{21}$$

where $\eta$ is the bulk dynamic viscosity, $K$ is the consistency, a function of the crystal volume fraction $\varphi$ and temperature $T$, whereas the yield stress ($\dot{\gamma}$) and flow index ($\tau_0$) are both functions of $\varphi$. More details about this bulk viscosity model can be found in Mader et al.[28]. In the numerical model, this viscosity formulation was regularized using the Papanastasiou's method. A source-based enthalpy method[66] was used to model the liquid–solid phase change (i.e., the crystallization)[27] by ensuring that the relationship between temperature and the liquid fraction $f$ showed in Supplementary Fig. 3 and discussed in the section entitled "Parametrization and validation of the thermal model" was satisfied. The computational mesh is composed of 54,000 cells. This mesh size was selected after a systematic study of the dependence of the results on mesh size[61]. Code validations are reported and extensively discussed in several previous heat and fluid flow studies[61,65].

On the other hand, the static case was modeled by a conductive heat transfer model, where the non-dimensional equation writes:

$$\frac{\partial T^*}{\partial \tau} = \nabla^2 T^* - \frac{1}{\mathrm{Ste}} \frac{\partial f}{\partial \tau}. \tag{22}$$

In this equation, the non-dimensional time is $\tau = t\alpha/L^2$. This equation was solved by a finite volume method in a structured grid coupled with an algorithm of phase change. Furthermore, the relation between $T^*$ and $f$ has been solved in a similar to the dynamic case. As in the dynamic case, we considered a two-dimensional computational domain and the same thermophysical properties. However, to model an open magmatic system refilled by a new parental magma and experiencing an incremental growth by addition of successive sills, the computational domain was periodically extended from below by the addition of new rectangular blocks of computational cells carrying initial magma properties and conditions (Fig. 8g–j). The growth is operated at different rates until the final size of the chamber is reached, which is equivalent to the size of the closed system (Fig. 8a).

**Parametrization and validation of the thermal model.** In the following we develop the parametrization for the thermal model. It allowed us to calibrate the simulation for different values of pressure, initial water content, and chemical composition of the parental magma that are relevant for the study of the fate of arc magmas. In detail, the first step to solve Eqs. (12), (13), and (14) consists in the definition of the dimensionless temperature.

For each experimental set (i.e., a group of experiments characterized by the same conditions of pressure, starting material and initial water content), the initial temperature $T_r$ was fixed at the liquidus temperature (Supplemetary Table 1). We assume that the crystal-bearing magma is surrounded by rocks characterized by a temperature at the far-field ($T_\infty$) reported in Supplemetary Table 1[27].

This configuration allowed us to study the evolution of the system from the liquidus to the attainment of a crystal volume fraction ($\varphi$) of ~0.5–0.55, a value close to the maximum packing fraction of a system where the crystals are characterized by aspect ratios ($r_p$) ranging from 1 to 3[28]. Close to the maximum packing fraction, the average viscosity of the magmatic system is several orders of magnitude larger than its initial value and large-scale convective motions are strongly inhibited[28,37]. Perturbations to the global behavior of the system and small scale motions due to, for example, crystal settling[67], a non-homogeneous distribution of crystals[29], or melt and fluid buoyancy[14] relative to the solid phase, are still possible but not accounted for in the model.

The initial thermophysical properties of the system are given in Table 1. Using the reference values given in Table 1, an aspect ratio ($r_p$) for the crystallizing assemblage equal to 2, and temperature values for $T_r$ and $T_\infty$ reported in Supplementary Table 1, the temperature difference $\Delta T_r$ ranges from 105 to 190 °C. Stephan and Prandtl numbers are in the ranges 0.30–0.54 and 550–3650, respectively. Numerical simulations have been performed for representative Stephan and Prandtl numbers of 0.43 and 2577, respectively. The latent heat of crystallization ($L_m$) is taken equal to $3.5 \times 10^5$ J kg$^{-1}$ in accordance with Annen et al.[41] and the viscosity of the melt phase is calculated using the method proposed by Giordano et al.[39]. The selected values strictly describe the physical conditions of the experiment reported by Nandedkar et al.[30]. Moreover, they can be assumed as proxies for the whole range of investigated thermophysical conditions. Indeed, at high Pr numbers, there is a negligible dependence of the flow and heat transfer on Pr[68]. This relatively high value of Prandtl number indicates that at the beginning of the cooling process, the diffusion of momentum by viscous action is predominant over the diffusion of heat inside the magmatic chamber. The value of the Stefan number just below one confirms the relative importance of the latent heat transfer in comparison with the sensible heat transfer. We performed a set of simulations to investigate the effect of the Ste number variations in the range 0.30–0.54, obtaining timescales estimation within 19% of those obtained at Ste equal to 0.43, as indicated in Supplementary Fig. 4.

To provide a solution for the latent heat term in Eq. (14), we defined an empirical relation between the melt fraction ($f$) and the dimensionless temperature ($T^*$). This relation is based on the petrological data set described in the section titled "Petrological data set" (Supplementary Fig. 3). Results from experimental data have been modeled using both a linear relation (reported in purple on Supplementary Fig. 3) and a polynomial curve (reported in yellow on Supplementary Fig. 3). Since the use of the two different models provide similar results (Supplementary Fig. 5), only the results obtained from the linear relationship between $f$ and $T^*$ are reported. In practice, the slope of the line reported in Supplementary Fig. 3 provides an estimation of $df/dT^*$ at each $T^*$ value.

Also, the empirical relationship linking the melt fraction ($f$) and the dimensionless temperature ($T^*$) reported in Supplementary Fig. 3 allowed us to constrain the rheological behavior of the system for the simulation of the dynamic configuration. In these cases, the rheological behavior of the system is modeled applying the Herschel–Bulkley formulation[28]. A wide range of thermodynamical conditions are investigated. In detail, we defined three buoyancy-driven convective systems characterized by different Rayleigh numbers (Ra) of $10^7$ (dynamic case 1, DC-1), $10^8$ (dynamic case 2, DC-2), and $10^9$ (dynamic case 3, DC-3)[67].

The parameter $\beta$ in the Eq. (19) is assumed as a constant and varied in accordance with Ra variations. This choice is in agreement with the linear parametrization reported in Supplementary Fig. 3, and it assumes that relative density variations in the melt phase and the crystal assemblage do not significantly change the value of the parameter $\beta$ during the evolution of the system.

To deal with the complex behavior of the studied system, we consider a non-linear model coupling the flow induced by buoyancy forces, resulting from the thermal expansion of the melt, the heat transfer, the phase change (i.e., formation of crystals), and the non-Newtonian rheological behavior of the fluid. The latter is a function of shear, temperature, and crystal volume fraction $\varphi$, accounted for by a Herschel–Bulkley model[27].

Figure 9 displays the temperature field of the system at four dimensionless times $\tau^* = 10$, 25, 50, and 100 (corresponding to $\tau = 3.88 \times 10^{-3}$, $9.7 \times 10^{-3}$, $1.94 \times 10^{-2}$, and $3.88 \times 10^{-2}$). Convective motions generate upside-down mushroom-like thermal plumes. These plumes originate at the roof of the system and progressively descend to the floor of the magma chamber. At low Ra (i.e., $10^7$), the location of consecutive plumes is almost regular; however, at higher Ra, the development of new plumes becomes irregular. Figure 9 also shows that the frequency of plume formation decreases with time, leading to the generation of a region of thermal stratification on the floor of the magma chamber.

The observed variations in the temperature fields at different Ra are due to different evolutions of the velocity field in the system. Supplementary Fig. 6 shows the velocity field for the same configurations and times shown in Fig. 9. As reported in Supplementary Fig. 6, the module of the velocity field varies by about an order of magnitude between $Ra = 10^7$ and $Ra = 10^9$ from the early stages of the simulation (e.g., $\tau^* = 10$). The observed increase in the vigor of the convection at growing Ra results in an increase of thermal mixing, leading to a faster decrease of the mean temperature of the system (Supplementary Fig. 6).

The final step to parametrize the evolution of $\Delta H_2O$ over the non-dimensional time consists in the transformation of the volume fraction $f$ utilized in the simulations to a mass fraction $X_{melt}$ (Eq. (16)). The obtained $X_{melt}$ values are then introduced in Eqs. (10) and (11) to obtain $\Delta H_2O$. The reported results are developed adopting $R_\rho$ value equal to 0.86, corresponding, for example, to a melt density ($\rho_{melt}$) equal to 2750 kg m$^{-3}$ and an average density of the crystal assemblage ($\rho_{cryst}$) equal to 3200 kg m$^{-3}$. The values for the densities of melt and crystals are in agreement to those characterizing the adopted petrological data set[30–33]. A reduction of the $R_\rho$ parameter of 0.07 is consistent with a decrease of the melt density ($\rho_{melt}$) of 250 kg m$^{-3}$. Moreover, a variation of the $R_\rho$ parameter of 0.07 will result in a change of the estimated timescales, calculated at $\Delta H_2O$ equal to 2.0, no larger than 15%.

**Melt migration velocities**. Magma ascent velocities are estimated using two different approaches, based respectively on the buoyancy-driven[43,44] and magma-driven[45,46] ascending dikes theory. With the former approach, the dike is assumed to be isolated and its propagation is due to buoyancy forces only. Indeed, using the elastostatic theory, Weertman[69] demonstrates that buoyancy forces are sufficient to allow vertical fractures to propagate upward, as soon as they are longer than a critical length[43]. In the latter, propagation is driven not only by buoyancy forces, but also by the excess pressure of a reservoir connected to the dike. Therefore, due to the force applied by the excess pressure, even dikes smaller than the critical length can propagate toward the surface. The computation of magma ascent velocities is disjointed from the thermal numerical simulations, allowing us a more comprehensive description of the fate of water-rich melts in subduction zones.

Following Dahm (2000)[43], the propagation velocity ($v$) of a buoyancy-driven ascending dike in the crust characterized by a height $2a$ and an average width $2h$ can be estimated using the following equation[43]:

$$v_{0,\text{Dahm}} = \frac{\Delta P_{\text{visc}}}{3\eta \int_{-a}^{+a} \tilde{h}^{-2}(z)\,dz}, \qquad (23)$$

where $\eta$ is the dynamic viscosity; $\Delta P_{\text{visc}}$ and $\tilde{h}(z)$ are defined as follows:

$$\Delta P_{\text{visc}} = 2a\Delta\rho g\left(1 - f^*\frac{K_c}{\sqrt{\pi}\Delta\rho\,g\,a^{3/2}}\right), \qquad (24)$$

$$\tilde{h}(z) = h(z) + \left[\frac{(1-\nu)K_c}{2G}\sqrt{\frac{a}{\pi}}\right]D. \qquad (25)$$

In Eqs. (24) and (25), $\Delta\rho$ is the density difference between crustal rocks and the rising melt, $g$ is the modulus of the gravitational acceleration, and $K_c$ is the fracture toughness. Furthermore, $G$ is the shear modulus, $\nu$ is the Poisson's ratio, and $D$ is an experimentally estimated dimensionless thickness, which it is assumed to be in the range 0.05–0.1 (in agreement with Dahm[43]). The upper and the lower limits of the blue and red areas reported in Fig. 6a, b correspond to a $D$ value equal to 0.05 and 0.1, respectively. The $f^*$ constant is a geometrical factor equal to 1 and $\pi/4$ in two and three dimensions, respectively. Finally, the half width of the dike $h$ at the depth $z$ is defined as follows[43,44]:

$$h(z) = \left[\frac{(1-\nu)K_c}{2G}\sqrt{\frac{a}{\pi}}\right]\left(1+\frac{z}{a}\right)\sqrt{1-\left(\frac{z}{a}\right)^2}. \qquad (26)$$

Crust rheological parameters are reported in Supplementary Table 3[70].

The average flow velocity estimated by Rubin[45], obtained adopting the magma-driven approach, has a similar formulation compared to that reported in Eq. (23):

$$v_{0,\text{Rubin}} = \frac{\Delta P_{\text{excess}}}{3\eta}\frac{(2\hat{h})^2}{2a}, \qquad (27)$$

where $\Delta P_{\text{excess}}$ is the uniform excess pressure and the average half thickness $\hat{h}$ is calculated as:

$$\hat{h} = \frac{\Delta P_{\text{excess}}}{\eta/(1-\nu)}a. \qquad (28)$$

The estimated ascent velocities reported previously are computed assuming an incompressible magma. However, the presence of volatiles in the magmatic mixture increase the ascent velocity, since, due to volatile exsolution, the density of the mixture decreases with pressure, increasing, as a result, the buoyancy forces. Taisne and Jaupart[46] have estimated the increase of the ascent velocity due to the presence of exsolved volatile phases using the following relation:

$$v = v_0\frac{\rho_{\text{melt}}}{\rho}\left[\frac{\rho(\rho_s - \rho)}{\rho_{\text{melt}}(\rho_s - \rho_{\text{melt}})}\right]^{1/3}. \qquad (29)$$

In the previous equation, $\rho_{\text{melt}}$ is the bubble-free melt density, $\rho_s$ is the density of the surrounding rocks, while $\rho$ is the density of the mixture calculated as:

$$\rho = (1 - \alpha_{\text{gas}})\rho_{\text{melt}} + \alpha_{\text{gas}}\rho_{\text{gas}}, \qquad (30)$$

where $\alpha_{\text{gas}}$ and $\rho_{\text{gas}}$ are, respectively, the volume fraction and the density of the exsolved gas mixture. In our computations, the density of the bubble-free melt and of the surrounding rocks are assumed constant, while $H_2O$ and $CO_2$ densities vary accordingly to the ideal gas law. The exsolved volatile contents and the relative proportion of exsolved $H_2O$ and $CO_2$ are calculated at different pressures using the mixture solubility law proposed by Ghiorso and Gualda[58].

Finally, the critical velocity $v_c$, defined as the minimum velocity at which the magma is able to arrive to the Earth surface, is defined as[47]:

$$v_c = \frac{H}{t_s}, \qquad (31)$$

where $H$ (25 and 35 km in our case) is the initial depth of the rising magma and $t_s$ is the solidification time defined as:

$$t_s = \frac{h^2}{4\alpha_r\lambda^2}, \qquad (32)$$

where $\alpha_r$ is the thermal diffusivity of the host rocks and $\lambda$ is given by the transcendental equation and fixed to a value of 0.54[47].

**Data availability**. The raw results of numerical simulations are available at the following repositories: https://goo.gl/GNgWbo and https://goo.gl/9CrBZt. Other data supporting the findings of this study are available as Supplementary Material and from the corresponding authors upon request.

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

## Acknowledgements

We wish to acknowledge the European Research Council for the Consolidator grant CHRONOS (612776) (D.P.). We were granted access to the HPC resources of MCIA Bordeaux (France). We acknowledge R. Astbury for proofreading of the manuscript.

## Author contributions

M.P. conceived and supervised the study. M.P., L.S., and D.P. investigated the thermo-dynamical evolution of the studied systems and developed the $H_2O$–$CO_2$ solubility computations. M.P., K.E.O., and Y.L.G. constrained the initial and boundary conditions for the thermal model. K.E.O. and Y.L.G. performed the numerical simulations for the thermal modeling. M.P., L.S., and G.L.S. managed the estimation of magma ascent rates. All the authors contributed to the data interpretation and participated in the final version of the article.

## Additional information

**Competing interests:** The authors declare no competing financial interests.

