## [Peer Review File · Nature Communications]

Reviewers' comments:

Reviewer #1 (Remarks to the Author):

Nature Communications manuscript NCOMMS-17-10777

The manuscript « Timescales of water accumulation in magmas and implications for short warning times of explosive eruptions » by Petrelli et al. deals with numerical simulations of the timescales of water accumulation in the residual melt of a crystallizing mafic magma and the ascent. Two problems are elucidated:

- how long does a residual melt of a crystallizing magma need to get enriched to 6 wt.% of water? It is solved by modeling stepwise phase equilibrium experiments from a previous study by Nandedkar et al. (2014).
- what are the ascent rates of those kind of residual melts leading to eruption? It is determined by calculations based on the buoyancy of magma in a denser host.

In this form, the manuscript is well organized; the message is clear and the reading easy to follow, even for non-specialists. Due to my limited skills for numerical simulations, I focused my review on the scaling from experimental petrology to the magmatic processes. I hope other reviewers will comment on the calculation and their robustness. My review includes major comments that I think to be critical for the manuscript and I wish to see revised (in the manuscript or in a separate answer if judges useful or not by the authors) and minor comments.

Major comments

#1 All the results are based on a single petrological study by Nandedkar et al. 2014 (CMP). My opinion is that it is a very limited selection for the generalization that follows. I invite the authors to prove that this single study is legitimate and representative of mafic arc magmas, or to consider more than one petrological study (Sisson doi:10.1007/BF00283225, Grove doi:10.1007/s00410-003-0448-z, Andujar <https://doi.org/10.1093/petrology/egv016>, Pichavant doi:10.1007/s00410-007-0208-6, Melekhova doi:10.1038/ngeo1781, Botcharnikov <https://doi.org/10.1093/petrology/egn043...?>). The latter option may bring robustness, but also enlarge the range of the results (temperature difference from the liquidus at H₂O = 6 wt.%; viscosity calculations, ascent rates...) found by the authors.

#2 A limitation of the study to me is the "so simple" relation between the water content of the residual melt upon crystallization of the magma. If I understood correctly, the authors assume that the magma half-crystallized double its water concentration. However, looking at the experiments used by the authors (Nandedkar et al., 2014 CMP), there is something like 16.6% of amphibole at 1010°C (« RN8 »). Amphibole containing water, I invite the authors to address more carefully the mineralogy, or to show that this hydrous phase can be omitted but I don't think it could be. The water in hydrous phases may deviate the [H₂O = 6wt.%] to a melt fraction of almost 0.60, with large consequences on the viscosity of the magma and potentially on the following discussion.

#3 A related issue to #2 is the discrepancy between experimental results of Nandedkar et al. and the argumentation of the authors here to calculated water content and crystal fraction. At 1010°C, the authors expect 50% of crystals in the magma (L122-123), but the experiments of Nandedkar et al. produced only ~25% of crystals (RN8, Table 2, Nandedkar et al.). How to deal with this difference to say the calculations are robust?

#4 I think that the starting water content of the basalt should be discussed. All the study is based on a basalt containing 3 wt.% of water. But what difference would it make with a basalt with 1-2 wt.% (Basalts from the Cascades, Sisson & Grove, 1993, CMP), with 4-5% (as presented in Plank et al., 2013, EPSL) or with the primitive magmas of water rich andesites from Mount Shasta

(Krawczynski et al., 2012) (all these references in the review of Grove et al., 2012, Annual reviews, <http://annualreviews.org/doi/abs/10.1146/annurev-earth-042711-105310>). Again, in the case I am wrong or there is no significant difference, the authors need to justify their choice.

#5 From L130: I am surprised by the simplification of the melt migration velocity calculations. If I understand well, only the buoyancy forces are taken into consideration, is that correct? But from 25 km depth (L267), other variables should be considered: (i) the decrease temperature leads to crystallization and increase of the viscosity (used in Eq. 14); (ii) the pressure decrease results in water exsolution out of the melt (at depths shallower than 8-10 km), and a significant crystallization, along with a change in the density difference (both used in Eq. 14 and 15). The Prandtl Number (and others) will depend on those parameters (L203-205). This is partially discussed L136-137, but too briefly.

#6 Maybe it is out of the scope of the present study, but perhaps the authors may take the chance to discuss the replenishment of reservoir. In their calculations, the authors consider the reservoir as a close system. It should be specified somewhere in the manuscript. What if the reservoir is open and replenished (with the frequency to be determined/tested)?

Minor comments

L25 and L27: "in addition" used twice. Perhaps another formulation would improve the style.

L40: the paragraph starting with "Recent investigations on..." introduces the amount of water in deep magmas as the previous paragraph. I recommend continuing the same paragraph.

L51: "in spite of recent developments". Vague; which developments?

L52: "continues to be blurred". Is there any more positive way to write this idea?

L55 and in the abstract: "what are the timescales required to increase the water content ... to more than 6 wt.%" I don't understand this key question. Why more than 6wt.% ? This critical value / threshold has not been introduced (the introduction mention water concentration of 6-9 wt.% or > 8 wt.%, but not 6 wt.% as a threshold. I suggest the authors to define what this value consists in. I imagine it corresponds to a density difference between the host rock and the residual, water enriched melt leading to critical buoyancy forces, or a water saturation pressure (~2.5 kbar) corresponding to a critical (?) depth (~8 km) of many reservoirs prior to eruption... This particular value may either correspond to the assumption (L69) that 50% of crystallization may double the water content of the residual liquid. In that case, such a relation should be introduced so the reader can understand why 6.wt% (see also major comments #2).

L87-88: "the size of the magmatic mass" is an odd formulation. I'd rather see the size of the magmatic reservoir, for example.

L116-117: the viscosity of the residual melt is calculated after Giordano et al. 2008. Please specify that the pressure effect has been omitted. Also, the superscript reference following the unit is confusing (editorial issue perhaps).

L117-118: What is the crystal fraction of the 2-phase magma viscosity presented here?

It is calculated after Mader et al., taking into account essentially 1 atm. analogical and experimental works. Is it consistent with the empirical model of Laumonier et al., 2014 (Nature Communications – doi:10.1038/ncomms6607) based on 3 kbar experiments? I briefly calculate a relative viscosity of 0.7 log unit for 26% of crystals according to Nandekar et al.'s experiments. Considering a crystal fraction of 40% (I don't know what value was determined by the authors), the relative viscosity is then about 1.2 log unit (crystal aspect ratio of about 2.9 average). (see also comment #3).

L127: "...or rise directly to the Earth's surface"?

L129: the authors may refer to the supplementary information. It could be added after the reference to the Figure 3B.

L139: the unit of the water content is misspelled.

L143: The sentence "the ascent of water rich melts might occur at..." should be moved to introduction and developed.

L146: "... water rich magmas can reach the Earth's surface in short timescales". The second point

of this study is the quantification of those timescales. If short here means 0.15 to 1.2 m/s as found by the authors (L134), then I recommend linking the paragraph so as to say it once only, and avoid the qualitative L146. If it is different, the authors should explain.

Also, I invite the authors to confront their findings with a larger literature. Only one reference (43) has been provided. (Myers et al., 2016: 0.3 to 1.5 m/s -

<https://doi.org/10.1016/j.epsl.2016.07.023>; Peslier et al., 2015: 0.2 to 25 m/s -

<https://doi.org/10.1016/j.gca.2015.01.030>; La Spina et al., 2016 -

<https://dx.doi.org/10.1038%2Fncomms13402>; Papale & Dobran, 1994 - DOI:

10.1029/93JB02972; Melnik & Sparks 2002 - doi: 10.1144/GSL.MEM.2002.021.01.07 ; ...

L147 and L156: The Earth's surface. The s is not necessary.

L163: "a magma close to the liquidus temperature". How much close?

L167-169: I don't understand the sentence "The rheology of the system at 0.7 GPa is linked to... at the same conditions of pressure and temperature and...". The viscosity, and thus the rheology, depends on the temperature (melt viscosity), which was included as a variable in the experiments of Nandedkar et al. Can the authors develop?

L177: What would be the typical Rayleigh number of a reservoir at natural conditions? Are 107 to 109 realistic?

L185++ (no line numbers): I did not see the definition of ∂f from Eq. 9 in the text.

L200: I would like to see a reference to Nandedkar et al.'s work (37) when citing their results.

Also, If the viscosity is calculated after Giordano et al., 2008, the reference should be in the text as well.

L212: (editorial) mind the reference affixed close to the -1 of the J.kg-1 unit.

L226: I recommend discussing shortly the effect to the crystal shape on the maximum packing fraction (Mader et al. 2013, or Laumonier et al., 2014).

L252: The authors may refer to Fig 6 to illustrate the last sentence.

L261: f was already used as the melt fraction in equation 9. Isn't it problematic?

L262: What is the physical meaning of the numbers h and e? Can it be explained if they have a meaning?

Equation 16: what is the experimentally estimated constant D? Where does it come from (reference)? How much is it here?

FIGURE 1: I guess the H₂O (wt.%) on the y axis represent the water dissolved in the melt and not in the "magmatic mass" since the magma is crystals + melt is set here to 3 wt.%. It should be corrected in caption and on the axis if I am right.

FIGURE 3: the reference to calculate viscosities (relative, bulk, melt) are missing in caption or in the legend.

FIGURE 4: I think an idea of the melt composition (e.g. SiO₂ content) would be useful to better understand the figure. Do the authors share the same idea? Without, I am not sure whether or not this figure includes the residual melt composition, while we are talking of a magma (basaltic parental composition) after 50% of crystallization.

Also, the 10km H₂O saturation value may be too low. According to the numerous references in Iacono-Marziano et al., 2012 (GCA, Figure 4c ; <https://doi.org/10.1016/j.gca.2012.08.035>), the water content under saturation conditions (10 km - 3.3 kbar) in mafic melts is likely to be ~7wt.% (Fig 7 of Iacono-Marziano et al., 2012).

FIGURE 6: What does the range 0 to 1 of T stand for? Is it 1010 to 1150°C?

FIGURE 7: when the velocity is not expressed with a log scale, what is its unit? m/s? s⁻¹? Is it similar to what we expected in magma reservoir (10⁻⁷ to 10⁻⁴ s⁻¹; e.g. Albertz et al., 2005 <http://dx.doi.org/10.1130/B25444.1>)?

May 22nd, 2017

Mickael LAUMONIER

Reviewer #2 (Remarks to the Author):

The manuscript presents a numerical investigation of the timescales required for water concentration increase in the residual melt of a magma body stored at mid-deep crustal levels, undergoing cooling-induced convection and crystallization. The results show timescales of years to tens of years to attain water concentrations and crystal contents that are considered to be sufficient to favour melt extraction and propagation towards the Earth's surface. Further analysis based on dyke propagation theory is used to add evaluations on possible timescales of magma transfer to the surface, and to provide recommendations for volcanic hazard forecasts.

I find the manuscript in principle interesting to a broad audience of geoscientists. At the same time, I think the pretended general applicability of the results and conclusions is seriously limited by some critical assumptions, that I report below. There is also a number of aspects that need to be better clarified in the manuscript, also reported.

As a general conclusion, I tend to consider this manuscript as a relevant contribution towards understanding the complex issues at its subject (evolution of buried magmatic bodies and their potential for eruption); although, the contribution is not fully convincing in terms of general applicability for the interpretation of real data. That is a serious limit, in my perception, to its potential of impacting a broad, heterogeneous audience like that of Nature Communications.

General comments

The manuscript aims at deriving general conclusions on the evolution of magmatic bodies stored at mid-crustal level; however, some assumptions seem to strongly limit the possibility of generalization:

1. The timescales for the evolution of large (km-size) magmatic bodies are computed by assuming that the body remains isolated from further magmatic inputs for thousands or tens of thousands years. That's hardly the case when compared with the body of geochemical and petrologic data on real magmas, that nearly invariably suggest periodic rejuvenation of magma chambers by deeper, hotter, less chemically evolved, volatile-rich magmas occurring over periods of time much lower than the analysed time-scales. Such periodic inputs of magma would deeply disrupt the computed temperature, chemical, crystallization and water accumulation patterns. In other words, the considered conditions seem to refer to very extreme cases, certainly worth of being studied, but hardly representative of the expected conditions and evolutions. That's not commented nor even mentioned in the manuscript (on the contrary, Fig. 4, that generalises the results, show deep magma bodies that are open at their bottom – as they are expected to be).

2. The conditions for the simulations are selected with reference to a suite of petrologic experiments, fully quoted, that refer to a specific pressure (0.7 GPa) and initial water content (3 wt%). The experimental results are heavily controlling the simulated dynamics: in fact, the authors link any evolving quantity (crystal content, water concentration in the melt, etc.) and any evolving property (viscosity, probably density although not clearly specified, etc.) to temperature through simple functions interpolating the experimental results. That's OK, but it strongly limits the possibility of generalizations to exactly those conditions that were used for the experiments; what about the evolution of a magma chamber stored at a lower/higher depth, and/or carrying a larger/lower initial water content? Such limitations are not discussed; on the contrary, the discussion and conclusions are developed as if the results were general and applicable under many different conditions.

More specific comments

- The discussion about water concentration in magmas stored below arc volcanoes (lines 30-50), and the corresponding conditions adopted for running and stopping the simulations, may be further clarified. It is reported (line 34) that a global average for arc magmas is 3.9 ± 0.4 wt% H₂O (it should be clarified that such an amount refers to mafic magmas, as from the quoted reference); however, the numerical simulations assume a significantly lower value of 3.0 wt% H₂O (I understand, in order to benefit from existing experiments made with such a water content). How large (or how small) is the approximation introduced, considering that 1 wt% more H₂O may have substantial effects on the liquidus temperature and the crystallization paths, and ultimately affect the computed timescales?
- On the same issue: it is said that recent investigation points to magmatic bodies at mid-crustal level carrying up to 9 wt% H₂O; however, the presented simulations stop at 6 wt% H₂O, where the viscosity of the melt-crystal mixture becomes exceedingly large, and before the melt becomes saturated with water and starts exsolving it. I understand that the very large water contents reported in the literature either refer to different pressure conditions and/or melt compositions, or they include substantial water stored in a gas phase; in any case, the results and discussion here do not seem to place such reported large water contents in a consistent picture.
- Line 69, and caption to Fig. 1: please clarify that the increase in water concentration pertains to the melt phase, not to the "magmatic mass" that the reader easily associates to the multiphase magma.
- Equations 1-4: I suggest that an effort is made here to help the less expert reader understand, avoiding misinterpretations. In fact, Ra strongly depends of the size of the system (L in the equations); while the different equations are for constant Ra, L is taken as the independent variable, implying that other conditions vary substantially from point to point along each curve represented by each one of eqs. 1-4. In this respects, the example reported at lines 95-97 is largely misleading: the less expert reader is induced to conclude that a larger magma chamber implies longer timescales (that's exactly what is said at such lines), not necessarily understanding that the reported comparison referring to constant Ra implies that the overall conditions in the comparison must be largely different (they imply magmas with different properties and different thermal disequilibrium conditions).
- Line 128: the discussion on melt migration velocities is not fully clear. First, once again, the less expert reader would be facilitated if a statement is included, clearly explaining that these computations (and the whole section B of the Methods) do not pertain to the numerical simulations of the convection dynamics, but are instead performed separately from them. Second, it should be clearly stated that the model neglects pressure forces as a driver of dyke propagation (the words "buoyancy-driven" are correctly included, but further clarifying it – what the expert reader fully understands from the equations in section B of the Methods – will definitely help). Third, although the reader is referred to the quoted paper by T. Dahm where the theory is developed, it would be of help to explain here how buoyancy can cause dyke propagation in a fractured medium. In fact, if the rheology of the medium is brittle, one does not expect any buoyancy effects, especially over the short timescales of hours to days emerging from the computations.
- Fig. 3a. The figure shows the numerical results in terms of association between chemical evolution of the melt and water accumulated in the melt phase (blue line). Because the manuscript aims at a general perspective, it should be discussed here if such an association emerges from global observations and data.
- Fig. 3b. The figure shows that the longer a dyke, the lower the minimum velocity required for the magma to reach the surface. That looks counterintuitive, and it may be conveniently explained. In the same figure, the width of the blue and red bands is not explained: what variabilities are

assumed to draw them?

- Lines 155-156. Recast the sentence.
- Methods, section A. Some details of the computations performed remain unclear. First of all, the initial conditions with location of a thermal boundary (if there is one; maybe it is exclusively cooling from the margins of the magma body? Lines 216-218 are not fully exhaustive) should be clearly reported. Second, magma properties and their variations should be more clearly explained. I understand the experimental results from Nandedkar et al. (2014) are interpolated, taking T as a proxy for H₂O content in melt, melt composition, and fractions of crystals precipitated. Are there specific models employed to compute melt (and/or multiphase) density, Newtonian melt viscosity (the methods for computing non-Newtonian multiphase viscosities are well referenced), latent heat released upon crystal precipitation, etc.? Referring to Table II, are all of the quantities reported changing during the simulations, and how are those changes computed?
- Line 178: change "Raleigh" to "Rayleigh".
- Eqs. 7-9 imply (to the expert reader) that the model is homogeneous (no separate flow of the melt phase and the crystals). That may be specified in the text.
- Eq. 17. The quantity h defined here was introduced as half the dyke width at line 257. Please explain. I'd suggest (not necessary) that the last bracket in the equation is placed before the square root symbol.

Reviewer #3 (Remarks to the Author):

Comments keyed to line numbers on pdf.

General comment

One aspect that is missing from this study is the role of CO₂. Now there is no doubt that mass-wise H₂O is the dominant volatile constituent. However, the presence of CO₂ does affect the thermodynamic properties of magmas in ways beyond the facile idea that the abundance of CO₂ is low and hence it does not play much of a role. The point is that because CO₂ is so insoluble a 'vapor' or supercritical fluid phase will virtually ALWAYS be [present, although at a small mass fraction of fluid. However, because H₂O is more soluble in CO₂ than in a melt, H₂O that otherwise would not be present in a fluid phase can be partitioned from melt into the CO₂ vapor. So the quantitative importance of this phenomenon should be examined. There are many mixed volatile models available to do some calculations to examine this effect. Papale has an older one and Ghirso et al a newer one. The authors can find the detailed refs.

Line 39: Or it could mean that fluid inclusions simply leak. In fact, see article by Adam Kent in MSA volume on inclusions. Many phases such as olivine and cpx leak volatiles. Hence the real meaning of measured water contents is not entirely clear is not east to interpret.

Line 54: some of these questions appear to have relatively simple, at least at broad brush, answer: Q1: Given initial H₂O total mass fraction in magma, as xllzn occurs H₂O builds up until it saturates! This can be modeled by MELTS for example. In fact, H₂O-CO₂ mixtures can be modeled quite well, nowadays so this is no great mystery. Q2: this is really a heat transfer problem. The main driver is the crystallization of nominally anhydrous phases. As that occurs the volatile content in residual under saturated melt build up until saturation that depends on P, T and bulk compo of the melt. So its really asking how quickly does the magma solidify. Q3: once melt is volatile-saturated, then it becomes quite compressible and so the bulk mixture density can fall rapidly upon ascent. This is a positive feedback and will drive the magma to accelerate.

Line 71: Not necessarily so. If the crystals are not homogeneously distributed there could be regions of 'clear melt'. The crystals will tend to settle but can be kept suspended by convective motions EXCEPT near a rigid momentum boundary layer within which sedimentation will take place. See Martin and Nokes JFM paper from late 1980's. The point is that the entire magma volume will eventually circulate through the bottom boundary layer where sedimentation may occur. This should be mentioned and discussed.

Line 78: you should give the relationship between dimensionless time and real time in the paper. I imagine this is $\kappa t/L^2$ or $\nu t/L^2$? Yes it is in the appendix but put it in the paper.

Line 84: what is the saturation water content for these examples. Obviously it would be the mean since P is constant and solubility is strongly dependent upon pressure. Once bubbles form then fractionation can occur, as bubbles will tend to rise. This can create 'gas caps' and strongly allow density stratification to develop that can greatly retard convective mixing by thermal convection. This should be discussed.

Line 160-250: since the density of the residual melt is strongly a function of the dissolved H₂O content of the melt that increases during crystal fractionation, what is the justification for not including the melt water content in the buoyancy term in the momentum equation as well as an (almost) hyperbolic equation expressing conservation of water? My point is that is of order unity for water in a typical melt whereas α , the isobaric temperature expansivity is of order 0.0001 or 0.00001. Hence the chemical buoyancy should not be neglected, should it? Of course, this makes the problem considerably more difficult especially because diffusion of heat is far more important than diffusion of water whereas dissolved water contributes perhaps more to buoyancy than does the temperature.

Line 255: I believe that the fracture propagation model used here treats magma as an incompressible fluid. Once volatile saturation occurs then the magmatic mixture is best treated as a compressible fluid. The propagation of cracks filled with a compressible fluid that undergoes ascent and decompression is a bit different than the fracture propagation of an incompressible fluid. There are some papers by Emanuel Detournay and co workers at University of Minnesota (engineering) that might be used to compare and contrast the propagation of cracks filled with compressible vs incompressible fluids. Magma of course can be both at various points along ascent path.

Reviewer #1

The manuscript « Timescales of water accumulation in magmas and implications for short warning times of explosive eruptions » by Petrelli et al. deals with numerical simulations of the timescales of water accumulation in the residual melt of a crystallizing mafic magma and the ascent. Two problems are elucidated:

- how long does a residual melt of a crystallizing magma need to get enriched to 6 wt.% of water? It is solved by modeling stepwise phase equilibrium experiments from a previous study by Nandedkar et al. (2014).
- what are the ascent rates of those kind of residual melts leading to eruption? It is determined by calculations based on the buoyancy of magma in a denser host.

In this form, the manuscript is well organized; the message is clear and the reading easy to follow, even for non-specialists. Due to my limited skills for numerical simulations, I focused my review on the scaling from experimental petrology to the magmatic processes. I hope other reviewers will comment on the calculation and their robustness. My review includes major comments that I think to be critical for the manuscript and I wish to see revised (in the manuscript or in a separate answer if judges useful or not by the authors) and minor comments.

We addressed all the main comments provided by the reviewer. In the following, we replied point-to-point to the reviewers' comments, and we explained the actions undertaken to resolve each issue.

Major comments

#1 All the results are based on a single petrological study by Nandedkar et al. 2014 (CMP). My opinion is that it is a very limited selection for the generalization that follows. I invite the authors to prove that this single study is legitimate and representative of mafic arc magmas, or to consider more than one petrological study (Sisson doi:10.1007/BF00283225, Grove doi:10.1007/s00410-003-0448-z, Andujar <https://doi.org/10.1093/petrology/egv016>, Pichavant doi:10.1007/s00410-007-0208-6, Melekhova doi:10.1038/ngo1781, Botcharnikov <https://doi.org/10.1093/petrology/egn043...?>). The latter option may bring robustness, but also enlarge the range of the results (temperature difference from the liquidus at H₂O = 6 wt.%; viscosity calculations, ascent rates...) found by the authors.

As suggested by the referee, we reviewed the existing literature concerning the petrology of volatile-bearing magmas in subduction zones to extended the study to more data sources concerning hydrous mafic arc magmas. In detail, in the revised version of the manuscript, we extended the petrological dataset utilized for the calibration of the thermal model. This action allowed the investigation of a larger pressure range (from only 0.7 GPa to an interval between 0.7 and 1.0 GPa) and different amounts of initial water contents (from the single value of ca. 3 wt.% to a range between 2 and 4.5 wt.%).

In detail, we utilized the results of 26 experiments based on the results reported by Nandedkar et al.³³, Melekhova et al.³⁴⁻³⁵, and Blatter et al.³⁶. Although the other studies suggested by the reviewer were carefully evaluated, we decided not to include them in the study since they do not provide a direct estimation of the H₂O concentration in the residual melt or they refer to pressure ranges that fall outside those investigated in the present study.

We thank the reviewer for the suggestion that allowed us to significantly increase the robustness of the model, enlarge the range of the obtained results and, moreover, improve overall significance of the manuscript (lines 321-338 and 468-555).

#2 A limitation of the study to me is the “so simple” relation between the water content of the residual melt upon crystallization of the magma. If I understood correctly, the authors assume that the magma half-crystallized double its water concentration. However, looking at the experiments used by the authors (Nandedkar et al., 2014 CMP), there is something like 16.6% of amphibole at 1010°C (« RN8 »). Amphibole containing water, I invite the authors to address more carefully the mineralogy, or to show that this hydrous phase can be omitted but I don't think it could be. The water in hydrous phases may deviate the [H₂O = 6wt.%] to a melt fraction of almost 0.60, with large consequences on the viscosity of the magma and potentially on the following discussion.

Following the suggestion provided by the reviewer, in the revised version of the manuscript we overcome the limitation of a “so simple” relation between the water content of the residual melt upon crystallization of the magma based on a single study. To do that, we move from the empirical relation proposed in the original version of the manuscript to a more robust framework based on the mass conservation law (lines 365-401). This operation allowed us the investigation of a wider range of initial water contents, to calibrate the model using a statistically robust dataset (26 experiments, significantly more than the 5 utilized in the original version of the manuscript), to evaluate the effects of the crystallization of hydrous phases (i.e. amphibole) and, also, to address the H₂O loss due to the reaching of the saturation of a CO₂- and H₂O-bearing system. We discussed in detail all these points in the “results” section (lines 90-154) and, also, in method section of the revised version of the manuscript (lines 321-338 and 468-555). In addition, following the suggestion provided by the reviewer, we carefully addressed the mineralogy of the system accounting for the presence of hydrous phases in the system. We thank the reviewer for this suggestion and we'd like to stress that the results presented in the original version of the manuscript are not affected significantly by the new calibration (since the original calibration was just a specific case that is completely consistent with the new formulation).

#3 A related issue to #2 is the discrepancy between experimental results of Nandedkar et al. and the argumentation of the authors here to calculated water content and crystal fraction. At 1010°C, the authors expect 50% of crystals in the magma (L122-123), but the experiments of Nandedkar et al. produced only ~25% of crystals (RN8, Table 2, Nandedkar et al.). How to deal with this difference to say the calculations are robust?

The reviewer is right when he states that RN8 experiment produced about ~25% of crystals. However, we would like to stress that RN8 represents an intermediate step of a series of experiments aimed to reproduce the process of fractional crystallization. This means that the amount of crystals produced in each single step must be added to those produced in the previous steps, after a normalization based on mass conservation (please refer to Nandedkar et al.³³ for further details). Therefore, at 1010°C the system contains 44.8% crystals (please refer to table 2 and figure 4 reported by Nandedkar et al.³³).

#4 I think that the starting water content of the basalt should be discussed. All the study is based on a basalt containing 3 wt.% of water. But what difference would it make with a basalt with 1-2 wt.% (Basalts from the Cascades, Sisson & Grove, 1993, CMP), with 4-5% (as presented in Plank et al., 2013, EPSL) or with the primitive magmas of water rich andesites from Mount Shasta (Krawczynski et al., 2012) (all these references in the review of Grove et al., 2012, Annual reviews, <http://annualreviews.org/doi/abs/10.1146/annurev-earth-042711-105310>). Again, in the case I am wrong or there is no significant difference, the authors need to justify their choice.

As suggested by the reviewer, we expanded the range of initial water contents considered in the present study. To do that, in light of the comment provided by the reviewer, we reviewed the available literature and we developed a new parametrization addressing initial water contents ranging from ~2.0 to ~4.5 wt%. We discussed this point in the revised version of the manuscript (lines 321-338 and 468-555).

#5 From L130: I am surprised by the simplification of the melt migration velocity calculations. If I understand well, only the buoyancy forces are taken into consideration, is that correct? But from 25 km depth (L267), other variables should be considered: (i) the decrease temperature leads to crystallization and increase of the viscosity (used in Eq. 14); (ii) the pressure decrease results in water exsolution out of the melt (at depths shallower than 8-10 km), and a significant crystallization, along with a change in the density difference (both used in Eq. 14 and 15). The Prandtl Number (and others) will depend on those parameters (L203-205). This is partially discussed L136-137, but too briefly.

Following the suggestion provided by the reviewer, we significantly improved the parametrization, the modelling and the discussion concerning the melt migration velocity. In the revised version of the model we:

- 1) included an additional model based on the parametrization proposed by Rubin et al.⁴⁸ in which melt migration velocity is driven not only by buoyancy forces but also by the reservoir's excess pressure. We compared the obtained results with those of the model proposed in the original version of the manuscript (Daham⁴⁷; lines 240-260 and 558-605);
- 2) evaluated the effects of volatile exsolution and decompression driven crystallization on the melt migration velocity (lines 260-286);
- 3) improved the overall discussion about melt migration velocities (lines 240-305).

Results from this new modelling further support those reported in the original version of the manuscript indicating that water-rich melts can attain the Earth surface in timescales of the order of days or even hours, possibly triggering explosive eruptions with little warning time and devoid of geophysical signals (lines 297-305).

#6 Maybe it is out of the scope of the present study, but perhaps the authors may take the chance to discuss the replenishment of reservoir. In their calculations, the authors consider the reservoir as a close system. It should be specified somewhere in the manuscript. What if the reservoir is open and replenished (with the frequency to be determined/tested)?

We thank the reviewer for the suggestion and, to account for the replenishment process, we developed further numerical simulations addressing both a closed system and an open system periodically refilled by new parental magma (lines 308-319). Results are thoroughly discussed at lines 161-181 of the revised version of the manuscript.

Minor comments

L25 and L27: “in addition” used twice. Perhaps another formulation would improve the style.

We rephrased the sentence in accordance to the reviewer’s comment (line 33).

L40: the paragraph starting with “Recent investigations on...” introduces the amount of water in deep magmas as the previous paragraph. I recommend continuing the same paragraph.

Done, as suggested by the reviewer (line 47).

L51: “in spite of recent developments”. Vague; which developments?

We rephrased the sentence in accordance to the reviewer’s comment (lines 60-61).

L52: “continues to be blurred”. Is there any more positive way to write this idea?

We rephrased the sentence in accordance to the reviewer’s comment (lines 60-61).

L55 and in the abstract: “what are the timescales required to increase the water content ... to more than 6 wt.%?” I don’t understand this key question. Why more than 6wt.% ? This critical value / threshold has not been introduced (the introduction mention water concentration of 6-9 wt.% or > 8 wt.%, but not 6 wt.% as a threshold. I suggest the authors to define what this value consists in. I imagine it corresponds to a density difference between the host rock and the residual, water enriched melt leading to critical buoyancy forces, or a water saturation pressure (~2.5 kbar) corresponding to a critical (?) depth (~8 km) of many reservoirs prior to eruption... This particular value may either correspond to the assumption (L69) that 50% of crystallization may double the water content of the residual liquid. In that case, such a relation should be introduce so the reader can understand why 6.wt% (see also major comments #2).

Re-reading the manuscript in light of the reviewer’s comment, we realized that the sentence at L55 was not clear since we considered a threshold value of 6 wt.% without explicitly stating that it represented the water content in the system at the jamming conditions, i.e. when the extraction of the residual melt from the magmatic mush starts being statistically most probable (Dufek and Bachman³²). We reported this in the revised version of the manuscript (lines 81-84).

In addition, to address this comment, and also following the suggestion provided by the reviewer in the major comment #2, in the revised version of the manuscript we studied a wider range of initial water contents (H₂O ranging from 2.0 to 4.5 wt.%) that results in water concentrations on the residual melt, at the jamming conditions, up ~ 6-9 wt.%. We amended the text accordingly.

L87-88: “the size of the magmatic mass” is an odd formulation. I’d rather see the size of the magmatic reservoir, for example.

Done, as suggested by the reviewer (line 196).

L116-117: the viscosity of the residual melt is calculated after Giordano et al. 2008. Please specify that the pressure effect has been omitted. Also, the superscript reference following the unit is confusing (editorial issue perhaps).

Done, as suggested by the reviewer (caption of Fig. 5).

L117-118: What is the crystal fraction of the 2-phase magma viscosity presented here?

It is calculated after Mader et al., taking into account essentially 1 atm. analogical and experimental works. Is it consistent with the empirical model of Laumonier et al., 2014 (Nature Communications – doi:10.1038/ncomms6607) based on 3 kbar experiments? I briefly calculate a relative viscosity of 0.7 log unit for 26% of crystals according to Nandedkar et al.'s experiments. Considering a crystal fraction of 40% (I don't know what value was determined by the authors), the relative viscosity is then about 1.2 log unit (crystal aspect ratio of about 2.9 average). (see also comment #3).

Following the suggestion of the reviewer, in the revised version of the manuscript we reported the crystal fraction of the 2-phase magma viscosity presented at lines 117-118 and we discussed our results in light of the empirical model reported by Laumonier et al.⁴² (lines 232-235 and Fig.5).

L127: “...or rise directly to the Earth' surface”?

Corrected in agreement to suggestion of the reviewer. In detail, we modified the term “move” with the term “rise” (line 239).

L129: the authors may refer to the supplementary information. It could be added after the reference to the Figure 3B.

Done, as suggested by the reviewer (line 242).

L139: the unit of the water content is misspelled.

Corrected, as suggested by the reviewer (line 288).

L143: The sentence “the ascent of water rich melts might occur at...” should be moved to introduction and developed.

In agreement to the comment provided by the reviewer, we developed the concept reported at line 143 (original version of the manuscript) to the introduction of the revised version of the manuscript (lines 84-88).

L146: “... water rich magmas can reach the Earth's surface in short timescales”. The second point of this study is the quantification of those timescales. If short here means 0.15 to 1.2 m/s as found by the authors (L134), then I recommend linking the paragraph so as to say it once only, and avoid the qualitative L146. If it is different, the authors should explain.

Following the suggestion provided by the reviewer, in the revised version of the manuscript we avoided qualitative estimations explicitly stating the meaning of ‘short timescales’ (lines 294-296).

Also, I invite the authors to confront their findings with a larger literature. Only one reference (43) has been provided. (Myers et al., 2016: 0.3 to 1.5 m/s - <https://doi.org/10.1016/j.epsl.2016.07.023>; Peslier et al., 2015: 0.2 to 25 m/s - <https://doi.org/10.1016/j.gca.2015.01.030>; La Spina et al., 2016 - <https://dx.doi.org/10.1038/ncomms13402>; Papale & Dobran, 1994 - DOI: 10.1029/93JB02972; Melnik & Sparks 2002 - doi: 10.1144/GSL.MEM.2002.021.01.07 ; ...

In agreement with the suggestion provided by the reviewer, in the revised version of the manuscript we compared our results on magma ascent timescale to a much wider range of natural estimates (Papale and Dobran⁵²; La Spina et al.⁵⁶; Myers et al.⁵⁷; Peslier et al.⁵⁵; Demouchy et al.⁵³; Lloyd et al.⁵⁴; Ray et al.⁵⁸). We did not include Melnik and Sparks (2002) because the model accounts for shallow conduit flow dynamics (<5 km) and dome extrusion activity, departing from the aim of this study (lines 266-270).

L147 and L156: The Earth's surface. The s is not necessary.

We removed the s as suggested by the reviewer (line 286).

L163: "a magma close to the liquidus temperature". How much close?

We rephrased the sentence and we specified that we started at the liquids temperature (line 475).

L167-169: I don't understand the sentence "The rheology of the system at 0.7 GPa is linked to... at the same conditions of pressure and temperature and...". The viscosity, and thus the rheology, depends on the temperature (melt viscosity), which was included as a variable in the experiments of Nandedkar et al. Can the authors develop?

As suggested by the reviewer, we improved the description regarding the parametrization of the thermal numerical simulations with the aim to avoid any potential misunderstanding (lines 468-555).

L177: What would be the typical Rayleigh number of a reservoir at natural conditions? Are 10^7 to 10^9 realistic?

In the revised version of the manuscript, we supported the choice for the selected values of Ra referencing the work by Martin and Nokes⁷¹ (line 519).

L185++ (no line numbers): I did not see the definition of ∂f from Eq. 9 in the text.

As suggested by the reviewer, in the revised version of the manuscript we reported the definition f, (i.e., the melt volume fraction) just after the equation 14 (former equation; lines 408-409).

L200: I would like to see a reference to Nandedkar et al.'s work (37) when citing their results. Also, If the viscosity is calculated after Giordano et al., 2008, the reference should be in the text as well.

As suggested by the reviewer, in the revised version of the manuscript we quoted the works by Nandedkar et al.³³ and Giordano et al.⁴³ where missing (e.g., line 494-496).

L212: (editorial) mind the reference affixed close to the -1 of the J.kg-1 unit.

Corrected, as suggested by the reviewer (line 493).

L226: I recommend discussing shortly the effect to the crystal shape on the maximum packing fraction (Mader et al. 2013, or Laumonier et al., 2014).

As suggested by the reviewer, in the revised version of the manuscript we discussed the effect of crystal shapes on the maximum packing fraction (lines 221-229 and 489).

L252: The authors may refer to Fig 6 to illustrate the last sentence.

Done, as suggested by the reviewer (line 544).

L261: f was already used as the melt fraction in equation 9. Isn't it problematic?

The reviewer is right and to avoid any misunderstanding, we change f to f^* in equation 9 (equation 24 of the revised version of the manuscript).

L262: What is the physical meaning of the numbers h and e? Can it be explained if they have a meaning?

The parameter e is just an auxiliary term that allowed us to have a more compact form of Eq. (16) and (17). The parameter h , instead, represent the half width of the dike, as already written in the manuscript. In the revised version of the manuscript, we removed the parameter e and we rewrote the equations (16) and (17) accordingly (equations 25 and 26 of the revised version of the manuscript). The meaning of each parameter is now explicitly reported in the revised version of the manuscript (lines 558-605).

Equation 16: what is the experimentally estimated constant D ? Where does it come from (reference)? How much is it here?

In the revised version of the manuscript, we clarified the meaning of the parameter D . Also, we reported the values of D used in the present study and the reference supporting our choice (line 575-578).

FIGURE 1: I guess the H_2O (wt.%) on the y axis represent the water dissolved in the melt and not in the “magmatic mass” since the magma is crystals + melt is set here to 3 wt.%. It should be corrected in caption and on the axis if I am right.

The reviewer is right, and, in the revised version of the manuscript, we corrected both the axis and the caption of Fig. 1 (original version of the manuscript, now Fig. 3).

FIGURE 3: the reference to calculate viscosities (relative, bulk, melt) are missing in caption or in the legend.

We added the missing reference in the caption and legend of the Fig. 5 in the revised version of the manuscript (Fig. 3 of the original version of the manuscript) as suggested by the reviewer.

FIGURE 4: I think an idea of the melt composition (e.g. SiO_2 content) would be useful to better understand the figure. Do the authors share the same idea? Without, I am not sure whether or not this figure includes the residual melt composition, while we are talking of a magma (basaltic parental composition) after 50% of crystallization.

Fig. 7 (Fig. 4 in the original version of the manuscript) only refers to the melt phase. We modified the picture to avoid misunderstandings.

Also, the 10km H_2O saturation value may be too low. According to the numerous references in Iacono-Marziano et al., 2012 (GCA, Figure 4c ; <https://doi.org/10.1016/j.gca.2012.08.035>), the water content under saturation conditions (10 km - 3.3 kbar) in mafic melts is likely to be ~7wt.% (Fig 7 of Iacono-Marziano et al., 2012).

In the revised version of the manuscript, we discussed in detail the H_2O and CO_2 saturation conditions of the ascending, volatile-rich melt. Regarding the specific question of the reviewer, we recalculated the H_2O solubility for the studied system at 10 km using the model described by Ghiorso and Gualda⁵⁹ and we obtained a value of 7 wt.% for a CO_2 free system. The obtained value agrees with the one suggested by the reviewer. As a consequence, we corrected it in Fig. 7 (revised version of the manuscript, former Fig. 4).

FIGURE 6: What does the range 0 to 1 of T stand for? Is it 1010 to 1150°C?

The reviewer is right, we indicated with 1 is the initial temperature utilized in the simulation and with 0 the final one. We specified this point in the caption of Fig. 10 (revised version of the manuscript, former Fig. 6).

FIGURE 7: when the velocity is not expressed with a log scale, what is its unit? m/s? s⁻¹? Is it similar to what we expected in magma reservoir (10⁻⁷ to 10⁻⁴ s⁻¹; e.g. Albertz et al., 2005 <http://dx.doi.org/10.1130/B25444.1>)?

The velocity in Figure 7 is dimensionless and normalized as described in the method section. When it is non-dimensionless it has the units of a velocity, i.e. m/s. Regarding the second part of the comment, we believe that a discussion about the obtained velocities in magma chamber is outside the aim of the manuscript. As a consequence, we decided to do not add it to the manuscript. However, we are open to add it if the editor and the reviewer believe that it is an important point.

Reviewer #2:

The manuscript presents a numerical investigation of the timescales required for water concentration increase in the residual melt of a magma body stored at mid-deep crustal levels, undergoing cooling-induced convection and crystallization. The results show timescales of years to tens of years to attain water concentrations and crystal contents that are considered to be sufficient to favour melt extraction and propagation towards the Earth's surface. Further analysis based on dyke propagation theory is used to add evaluations on possible timescales of magma transfer to the surface, and to provide recommendations for volcanic hazard forecasts.

I find the manuscript in principle interesting to a broad audience of geoscientists. At the same time, I think the pretended general applicability of the results and conclusions is seriously limited by some critical assumptions, that I report below. There is also a number of aspects that need to be better clarified in the manuscript, also reported.

As a general conclusion, I tend to consider this manuscript as a relevant contribution towards understanding the complex issues at its subject (evolution of buried magmatic bodies and their potential for eruption); although, the contribution is not fully convincing in terms of general applicability for the interpretation of real data. That is a serious limit, in my perception, to its potential of impacting a broad, heterogeneous audience like that of Nature Communications.

Please see our replies below where we replied point-to-point to the reviewers' comments, and we explained the actions undertaken to resolve each issue.

General comments

The manuscript aims at deriving general conclusions on the evolution of magmatic bodies stored at mid-crustal level; however, some assumptions seem to strongly limit the possibility of generalization:

1. The timescales for the evolution of large (km-size) magmatic bodies are computed by assuming that the body remains isolated from further magmatic inputs for thousands or tens of thousands years. That's hardly the case when compared with the body of geochemical and petrologic data on real magmas, that nearly invariably suggest periodic rejuvenation of magma chambers by deeper, hotter, less chemically evolved, volatile-rich magmas occurring over periods of time much lower than the analysed time-scales. Such periodic inputs of magma would deeply disrupt the computed temperature, chemical, crystallization and water accumulation patterns. In other words, the considered conditions seem to refer to very extreme cases, certainly worth of being studied, but hardly representative of the expected conditions and evolutions. That's not commented nor even mentioned in the manuscript (on the contrary, Fig. 4, that generalises the results, show deep magma bodies that are open at their bottom – as they are expected to be).

To address the comment provided by the reviewer, we developed new intensive numerical simulations accounting for periodic inputs of new magmas in the system. In detail, we investigated different scenarios accounting for: 1) the cooling of an isolated magmatic system exchanging heat only by conduction; 2) the cooling of a magmatic system that is periodically refilled by new parental magmas; 3) the effects related to the development of convective dynamics within the

system. We discussed these points in the revised version of the manuscript (lines 161-181 and 308-319).

2. The conditions for the simulations are selected with reference to a suite of petrologic experiments, fully quoted, that refer to a specific pressure (0.7 GPa) and initial water content (3 wt%). The experimental results are heavily controlling the simulated dynamics: in fact, the authors link any evolving quantity (crystal content, water concentration in the melt, etc.) and any evolving property (viscosity, probably density although not clearly specified, etc.) to temperature through simple functions interpolating the experimental results. That's OK, but it strongly limits the possibility of generalizations to exactly those conditions that were used for the experiments; what about the evolution of a magma chamber stored at a lower/higher depth, and/or carrying a larger/lower initial water content? Such limitations are not discussed; on the contrary, the discussion and conclusions are developed as if the results were general and applicable under many different conditions.

We agree with the reviewer when he states, referring the parametrization utilized in the original version of the manuscript, that it "strongly limits the possibility of generalization" of the obtained results". To overcome this limitation, we decided to improve the conceptual model for the evolution of the magmatic system (lines 321-338), to define a general framework for the parametrization of the thermal numerical simulations (lines 468-555), and to expand the dataset adopted for the calibration to more experimental data (now 26 experiments, they were 5 in the original version of the manuscript; lines 321-338). In particular, the new petrological calibration covers a wider range of pressures (from a single pressure value of 0.7 GPa to an interval ranging between 0.7 and 1.0 GPa) and initial water contents (from a single value of ~ 3 wt.% to an interval between 2 and 4.5 wt.%). We believe that the new parametrization allows us to discuss, in general terms, the evolution on a magmatic system stored at mid- to deep- crustal levels in subduction zones.

More specific comments

- The discussion about water concentration in magmas stored below arc volcanoes (lines 30-50), and the corresponding conditions adopted for running and stopping the simulations, may be further clarified. It is reported (line 34) that a global average for arc magmas is 3.9 ± 0.4 wt% H₂O (it should be clarified that such an amount refers to mafic magmas, as from the quoted reference); however, the numerical simulations assume a significantly lower value of 3.0 wt% H₂O (I understand, in order to benefit from existing experiments made with such a water content). How large (or how small) is the approximation introduced, considering that 1 wt% more H₂O may have substantial effects on the liquidus temperature and the crystallization paths, and ultimately affect the computed timescales?

As suggested by the reviewer, in the revised version of the manuscript we clarified the discussion reported at lines 30-50 (original version of the manuscript). First, we explained that the value of 3.9 ± 0.4 wt% H₂O refers to the global average for arc mafic magmas (Plank et al.²⁵). Also, we performed a new parametrization accounting for initial water contents in the parental magma ranging from 2 to 4.5 wt.% (lines 39 and 71-88). All the new parametrization governing the thermal numerical simulations are discussed in detail in the revised version of the manuscript

(lines 468-555). As an example, we accounted for the influence of the petrology of the investigated systems on the thermo-dynamical parameters (e.g. Stephan and Prandtl numbers) used in the simulations, and ultimately on the computed timescales (lines 321-338 and 468-555).

- On the same issue: it is said that recent investigation points to magmatic bodies at mid-crustal level carrying up to 9 wt% H₂O; however, the presented simulations stop at 6 wt% H₂O, where the viscosity of the melt-crystal mixture becomes exceedingly large, and before the melt becomes saturated with water and starts exsolving it. I understand that the very large water contents reported in the literature either refer to different pressure conditions and/or melt compositions, or they include substantial water stored in a gas phase; in any case, the results and discussion here do not seem to place such reported large water contents in a consistent picture.

To address the comment provided by the reviewer, in the revised version of the manuscript, we extended the simulations to a wider range of initial water contents allowing us the attainment of final water contents in the residual melt up to ~ 8-9 wt.% (lines 71-88, 321-338, 365-401, and 468-555).

Also, we discussed in detail the saturation conditions of the system for H₂O and CO₂ providing a general and consistent picture of the investigated problem (lines 90-154).

- Line 69, and caption to Fig. 1: please clarify that the increase in water concentration pertains to the melt phase, not to the “magmatic mass” that the reader easily associates to the multiphase magma.

Done, as suggested by the reviewer.

- Equations 1-4: I suggest that an effort is made here to help the less expert reader understand, avoiding misinterpretations. In fact, Ra strongly depends of the size of the system (L in the equations); while the different equations are for constant Ra, L is taken as the independent variable, implying that other conditions vary substantially from point to point along each curve represented by each one of eqs. 1-4. In this respects, the example reported at lines 95-97 is largely misleading: the less expert reader is induced to conclude that a larger magma chamber implies longer timescales (that’s exactly what is said at such lines), not necessarily understanding that the reported comparison referring to constant Ra implies that the overall conditions in the comparison must be largely different (they imply magmas with different properties and different thermal disequilibrium conditions).

To address the comment provided by the reviewer and avoid misinterpretations, we removed the example reported at lines 95-97 (original version of the manuscript) and we carefully checked the text to make sure that all the parametrization and the scaling was clear to avoid any misunderstanding.

- Line 128: the discussion on melt migration velocities is not fully clear. First, once again, the less expert reader would be facilitated if a statement is included, clearly explaining that these computations (and the whole section B of the Methods) do not pertain to the numerical simulations of the convection dynamics, but are instead performed separately from them. Second, it should be clearly stated that the model neglects pressure forces as a driver of dyke propagation (the words “buoyancy-driven” are correctly included, but further clarifying it – what the expert reader fully understands from the equations in section B of the Methods – will definitely help). Third, although the reader is referred to the quoted paper by T. Dahm where the theory is

developed, it would be of help to explain here how buoyancy can cause dyke propagation in a fractured medium. In fact, if the rheology of the medium is brittle, one does not expect any buoyancy effects, especially over the short timescales of hours to days emerging from the computations.

To address the first part of the comment, in the revised version of the manuscript, we clearly reported that the computations about melt migration velocities are separated from the numerical simulations concerning the thermal evolution of the magmatic systems stored at mid- to deep-crustal levels (lines 566-568).

Regarding the second and the third part of the comment, in addition to the model of Dahm⁴⁷ which considers only buoyancy-driven forces, we introduced a second approach based on the “magma-driven” theory, illustrated by Rubin⁴⁸. Here, the propagation is driven not only by buoyancy forces but also by the excess pressure of a reservoir connected to the dike. We clarified and discussed in detail all the assumptions of the two adopted models in the main text and in the methods section (lines 240-286; 558-605).

Finally, we discussed the role of volatile exsolution on the ascent rates using the parametrization proposed by Taisne and Jaupart⁴⁹ (lines 271-286).

- Fig. 3a. The figure shows the numerical results in terms of association between chemical evolution of the melt and water accumulated in the melt phase (blue line). Because the manuscript aims at a general perspective, it should be discussed here if such an association emerges from global observations and data.

As suggested by the reviewer, in the revised version of the manuscript, we generalized the discussion concerning Fig. 5 (Fig. 3a of the original version of the manuscript). In detail, we expanded the discussion from a single pressure 0.7 GPa to a pressure range between 0.7 and 1.0 GPa. Also, we discussed the role crystal aspect ratios in the evolution of the viscosity of crystal-bearing magmas. Finally, we compared our model to that reported by Laumonier et al.⁴¹ (lines 221-239).

- Fig. 3b. The figure shows that the longer a dyke, the lower the minimum velocity required for the magma to reach the surface. That looks counterintuitive, and it may be conveniently explained. In the same figure, the width of the blue and red bands is not explained: what variabilities are assumed to draw them?

Observing the Fig3a (original version of the manuscript, now Fig. 6a and 6b) in light of the reviewer’s comment, we realized that it deserved a further explanation. As indicated in the method section, the minimum ascent velocity is inversely proportional to the solidification time, which, in turn, can be estimated as a quadratic function of the dike width (h in equation 25). In the adopted formulation, the parameter h increases with the dyke length ($2a$) and, as a consequence, the minimum velocity required for the magma to reach the surface decreases with the increase of the dyke width.

To address the second part of the comment, we reported in detail how the blue and red bands have been computed. In particular, they are related to the adoption of a range for the D parameter between 0.05 and 0.1 in agreement with Dahm⁴⁷. We specified this point in the revised version of the manuscript (lines 577-578).

- Lines 155-156. Recast the sentence.

Done, as suggested by the reviewer.

- Methods, section A. Some details of the computations performed remain unclear. First of all, the initial conditions with location of a thermal boundary (if there is one; maybe it is exclusively cooling from the margins of the magma body? Lines 216-218 are not fully exhaustive) should be clearly reported. Second, magma properties and their variations should be more clearly explained. I understand the experimental results from Nandedkar et al. (2014) are interpolated, taking T as a proxy for H₂O content in melt, melt composition, and fractions of crystals precipitated. Are there specific models employed to compute melt (and/or multiphase) density, Newtonian melt viscosity (the methods for computing non-Newtonian multiphase viscosities are well referenced), latent heat released upon crystal precipitation, etc.? Referring to Table II, are all of the quantities reported changing during the simulations, and how are those changes computed?

To address the comment provided by the reviewer we improved the description and the parametrization of the thermal model. In detail, we clarified all the parametrizations (e.g. density, viscosity, latent heat etc.) and we defined in more detail the initial and boundary conditions.

In the revised version of the manuscript, we firstly described the governing equations and the numerical model (lines 403-465). Then, we developed the parametrization of the model (lines 468-555) in agreement with the adopted petrological dataset based on 26 experiments (lines 321-338).

- Line 178: change “Raleigh” to “Rayleigh”.

Done, as suggested by the reviewer (line 518).

- Eqs. 7-9 imply (to the expert reader) that the model is homogeneous (no separate flow of the melt phase and the crystals). That may be specified in the text.

As suggested by the reviewer, we specified this point in the revised version of the manuscript (lines 405-407)

- Eq. 17. The quantity h defined here was introduced as half the dyke width at line 257. Please explain. I’d suggest (not necessary) that the last bracket in the equation is placed before the square root symbol.

The half width of the dike h was defined accordingly to Dahm⁴⁷ and Rivalta et al.⁵⁰. In order to highlight this, we have introduced in the text the proper references. We also modified the equation in accordance to the suggestion provided by the reviewer (lines 579-581).

Reviewer #3:

Comments keyed to line numbers on pdf.

General comment

One aspect that is missing from this study is the role of CO₂. Now there is no doubt that mass-wise H₂O is the dominant volatile constituent. However, the presence of CO₂ does affect the thermodynamic properties of magmas in ways beyond the facile idea that the abundance of CO₂ is low and hence it does not play much of a role. The point is that because CO₂ is so insoluble a 'vapor' or supercritical fluid phase will virtually ALWAYS be [present, although at a small mass fraction of fluid. However, because H₂O is more soluble in CO₂ than in a melt, H₂O that otherwise would not be present in a fluid phase can be partitioned from melt into the CO₂ vapor. So the quantitative importance of this phenomenon should be examined. There are many mixed volatile models available to do some calculations to examine this effect. Papale has an older one and Ghiroso et al a newer one. The authors can find the detailed refs.

To address the comment of the reviewer, in the revised version of the manuscript we discussed in detail the role of CO₂ in the studied magmatic systems (lines 90-154). Particularly, we applied the model proposed by Ghiroso and Gualda⁵⁹ (lines 340-350 and 352-363) to: 1) evaluate the saturation curves for H₂O and CO₂ at different pressures for the parental magma (lines 90-108); 2) study the evolution of the saturation curves for H₂O and CO₂ during the cooling (and crystallization) of the system (lines 108-113); 3) evaluate the combined evolution of CO₂ and H₂O in the residual melt during the cooling (and crystallization) of the system (lines 114-154); 4) evaluate the CO₂ and H₂O saturation condition during the ascending of water rich melts from mid-to deep-crustal levels to the surface (lines 271-286). We thank the reviewer for this comment that, in our opinion, allowed us to significantly improve the overall significance of the manuscript.

Line 39: Or it could mean that fluid inclusions simply leak. In fact, see article by Adam Kent in MSA volume on inclusions. Many phases such as olivine and cpx leak volatiles. Hence the real meaning of measured water contents is not entirely clear is not east to interpret.

We completely agree that the "real meaning" of measured water contents from melt/fluid inclusions is not entirely clear and is not east to interpret, in agreement with the comment provided by the reviewer. In fact, we consider them not reliable to derive the water content (see line 45-47).

Line 54: some of these questions appear to have relatively simple, at least at broad brush, answer: Q1: Given initial H₂O total mass fraction in magma, as xllzn occurs H₂O builds up until it saturates! This can be modeled by MELTS for example. In fact, H₂O-CO₂ mixtures can be modeled quite well, nowadays so this is no great mystery. Q2: this is really a heat transfer problem. The main driver is the crystallization of nominally anhydrous phases. As that occurs the volatile content in residual under saturated melt build up until saturation that depends on P, T and bulk compo of the melt. So its really asking how quickly does the magma solidify. Q3: once melt is volatile-saturated, then it becomes quite compressible and so the bulk mixture density can fall rapidly upon ascent. This is a positive feedback and will drive the magma to accelerate.

Re-reading the manuscript in light of the comment provided by the reviewer, we realised that the questions Q1, Q2, and Q3, which were posed to gently introduce the reader to the main objectives of the present study, were, in some cases, simplistic and potentially misleading. We fully quote the comment provided by the reviewer when he said "Q1: Given initial H₂O total mass fraction in

magma, as crystallization occurs H₂O builds up until it saturates”, “Q2: this is really a heat transfer problem” and “Q3: once melt is volatile-saturated, then it becomes quite compressible and so the bulk mixture density can fall rapidly upon ascent. This is a positive feedback and will drive the magma to accelerate”. In the revised version of the manuscript, we better introduced the main objectives of the present study (lines 60-88). Also, we addressed the role of volatile exsolution during the ascent of a volatile-rich residual melt (lines 271-286).

Line 71: Not necessarily so. If the crystals are not homogeneously distributed there could be regions of ‘clear melt’. The crystals will tend to settle but can be kept suspended by convective motions EXCEPT near a rigid momentum boundary layer within which sedimentation will take place. See Martin and Nokes JFM paper from late 1980’s. The point is that the entire magma volume will eventually circulate through the bottom boundary layer where sedimentation may occur. This should be mentioned and discussed.

To address the comment provided by the reviewer, in the revised version of the manuscript we rephrased the sentence at line 71 specifying that we stop our simulation at a crystal content close to the jamming conditions since they represent, in accordance with Dufek and Bachmann³², the opening of crystallinity windows where the melt segregation from the magmatic mush is most probable (lines 81-84). Also, we reported, in the method section, a paragraph where we specified that close to the maximum packing fraction, the average viscosity of the system is order of magnitude larger than its initial value inhibiting large scale motions. We also specified that locally, perturbation to the average behaviour of the system due to crystal settling (Martin and Nokes⁷⁰), a non-homogeneous distribution of crystals (Dufek and Bachmann³²) or melt and fluid buoyancy (Parmigiani et al.¹⁶) relative to the solid phase are still possible but not accounted in the model (lines 478-486).

Line 78: you should give the relationship between dimensionless time and real time in the paper. I imagine this is $\kappa t/L^2$ or $\nu t/L^2$? Yes it is in the appendix but put it in the paper.

As suggested by the reviewer, in the revised version of the manuscript, we introduced the relationships between dimensionless time (τ) and dimensional time (t) both in the first appearance of τ in the main text (line 160) and in the caption of Fig. 3 (revised version of the manuscript).

Line 84: what is the saturation water content for these examples. Obviously it would be the mean since P is constant and solubility is strongly dependent upon pressure. Once bubbles form then fractionation can occur, as bubbles will tend to rise. This can create ‘gas caps’ and strongly allow density stratification to develop that can greatly retard convective mixing by thermal convection. This should be discussed.

As suggested by the reviewer, in the revised version of the manuscript we reported a comprehensive discussion about the H₂O and CO₂ saturations conditions of the reported examples. During our investigations, we found that volatile exsolution does not play a significant role for the thermal evolution of a magmatic system at 0.7 and 1.0 GPa. In particular, the evolution of the water content (parametrized using ΔH_2O ; i.e., the ratio between the water content in the melt phase and its initial value) is weakly conditioned by the attainment of the saturation conditions. On the contrary, the concentration of carbon dioxide is buffered at the saturation levels. We discussed these points in the revised version of the manuscript (lines 108-154).

Line 160-250: since the density of the residual melt is strongly a function of the dissolved H₂O content of the melt that increases during crystal fractionation, what is the justification for not including the melt water content in the buoyancy term in the momentum equation as well as an (almost) hyperbolic equation expressing conservation of water? My point is that is of order unity for water in a typical melt whereas alpha, the isobaric temperature expansivity is of order 0.0001 or 0.00001. Hence the chemical buoyancy should not be neglected, should it? Of course, this makes the problem considerably more difficult especially because diffusion of heat is far more important than diffusion of water whereas dissolved water contributes perhaps more to buoyancy than does the temperature.

Re-reading the text in the light of the reviewer comment, we realized that, in the original version of the manuscript, the definition of the buoyancy forces governing the development of natural convection within the system was not completely clear leading to potential misunderstandings. In particular, within the scope of the proposed model, the expansion coefficient β reported in equation (19) is minded accounting for both the chemical variations within the system (i.e., density changes in the melt phases due to chemical changes also accounting for water and the crystallization of solid phases) and the thermal expansion. In particular, the crystals are considered as being carried by the magma without any settling or floating, including in the boundary layer. Thus, the bulk density of the magma is not only related to the density of the melt, which indeed experiences a change in its water content, but it is related to both the melt phase and crystals (i.e., the bulk amount of water in each computational domain is constant). Therefore, the effect of water content variations on the bulk density of the magma is much weaker than its impact on the sole melt phase. We clarified this point in the revised version of the manuscript explicitly indicating that the parameter β in equation (19; i.e., the definition of the Rayleigh number) is minded as an effective expansion coefficient (lines 430-433 and 520-524).

Line 255: I believe that the fracture propagation model used here treats magma as an incompressible fluid. Once volatile saturation occurs then the magmatic mixture is best treated as a compressible fluid. The propagation of cracks filled with a compressible fluid that undergoes ascent and decompression is a bit different than the fracture propagation of an incompressible fluid. There are some papers by Emanuel Detournay and co workers at University of Minnesota (engineering) that might be used to compare and contrast the propagation of cracks filled with compressible vs incompressible fluids. Magma of course can be both at various points along ascent path.

The reviewer is right, in the original version of the manuscript the ascending magma was treated as an incompressible fluid. In the revised manuscript, we made an effort to account for the compressible behaviour on the magma due to presence of exsolved volatiles using the formulation reported in Taisne and Jaupart⁴⁹ (lines 586-599). We discussed in this point in the revised version of the manuscript (lines 260-270).

We decided to discard the model proposed by Detournay and co-workers (e.g. Lhomme et al., 2005; Detournay, 2006) since they are strongly focused on pit mechanics, not directly applicable to the problem studied in the present manuscript.

Lhomme, T., Detournay, E., & Jeffrey, R. (2005, March). Effect of fluid compressibility and borehole radius on the propagation of a fluid-driven fracture. In Proceedings of 11th International Conference on Fracture. Turin, Italy (Vol. 232).

Detournay, E. (2016). Mechanics of hydraulic fractures. Annual Review of Fluid Mechanics, 48, 311-339.

REVIEWERS' COMMENTS:

Reviewer #1 (Remarks to the Author):

NCOMMS-17-10777A, round #2

Overall, the authors made the effort to consider my comments and subsequently modify the text or explain their choices concerning the petrological constraints used in their numerical models. The previous limitations are now well argued. Therefore, I would recommend the manuscript as publishable in *Nature Communications*. Few remarks, though:

Fig. 1 b: I would like to see the nature (or simply the SiO₂ content) of the residual melts, and of the parental melt.

Define in the text the crystal fraction « ϕ » as it appears for the first time.

L121: replace Fig 1c by Fig 1d.

L130: the melt fraction is defined here as X_{melt} . It becomes confusing with the melt volume fraction f (L408). I would recommend to use a single variable through the entire manuscript (crystal fraction, melt fraction, either in mass or volume) to simplify the manuscript.

L227: The model of Laumonier et al., 2014 (*NatComm*) is valid up to crystal fraction of ~ 0.6 . I would suggest to draw the "brown curves" of Fig. 5 in dashed.

L337: Wrong ! Amphibole appears at crystal fraction = 0.25 in RN8 of Nandedkar et al 2014).

Discussion about the magma strain rate in reservoir is, to me, not important for the topic of such manuscript (as mentioned in the review letter).

M Laumonier,
November 9th 2017

Reviewer #2 (Remarks to the Author):

I had appreciated the first version of the manuscript, and raised a number of points that could improve it to the level required for a *Nature Geoscience* publication. I understand the authors have put substantial efforts in taking into account all of the comments and suggestions they received from all reviewers. That includes many new calculations referring to additional conditions, in order to demonstrate general applicability of their results. However, sincerely, I must conclude that the final result is much worse than the original. The ms has literally exploded, with 7 pages and 3 figures more (excluding methods), and a largely exceeding amount of details that are not suited for the main text - even an expert gets lost in too many numbers, conditions, exceptions, possibilities, etc.

I still believe the ms contains interesting and relevant results that may deserve publication in *NG*. My suggestion is that the authors try to get back to the original format, adding the information required in the simplest possible form, and moving the details on the many conditions explored and the many calculations done to the methods. The main text should be accessible and interesting for a broad audience; in the present format, it is not.

Reviewer #3 (Remarks to the Author):

the authors have responded to my comments to my satisfaction

Reviewer #1
NCOMMS-17-10777A, round #2

Overall, the authors made the effort to consider my comments and subsequently modify the text or explain their choices concerning the petrological constraints used in their numerical models. The previous limitations are now well argued.

We thank the reviewer for the comment.

Therefore, I would recommend the manuscript as publishable in *Nature Communications*. Few remarks, though:

Fig. 1 b: I would like to see the nature (or simply the SiO₂ content) of the residual melts, and of the parental melt.

As suggested by the reviewer, we reported the SiO₂ content of the parental and residual melts in the caption of Fig. 1.

Define in the text the crystal fraction « phi » as it appears for the first time.

All the parameters have been defined at first appearance, as suggested by the reviewer.

L121: replace Fig 1c by Fig 1d.

We adjusted the links to Fig 1c by Fig 1d in the sentence at L121 in order to avoid misunderstandings (lines 122-123).

L130: the melt fraction is defined here as X_{melt} . It becomes confusing with the melt volume fraction f (L408). I would recommend to use a single variable through the entire manuscript (crystal fraction, melt fraction, either in mass or volume) to simplify the manuscript.

All the results and discussion in the main text are already reported in term of X_{melt} . In the methods section, the use of volume melt fraction is required by eq. 14 (i.e. one of the Navier-Stokes equations). Including X_{melt} in this equation will make it much more complex and difficult to read. Anyway, the relation between X_{melt} and volume melt fraction is reported in eq. 16. In our opinion, this makes the paper more legible and we prefer to keep this formulation.

L227: The model of Laumonier et al., 2014 (*NatComm*) is valid up to crystal fraction of ~ 0.6 . I would suggest to draw the “brown curves” of Fig. 5 in dashed.

In the revised version of the manuscript, we reported that the model of Laumonier et al. (2014) is valid up to crystal fraction of ~ 0.6 and we draw the “brown curves” of Fig. 5 in dashed, as suggested by the reviewer.

L337: Wrong! Amphibole appears at crystal fraction = 0.25 in RN8 of Nandedkar et al 2014).

We smoothed this statement reporting that hornblende is “mostly appearing in experiments at crystal contents larger than 50%.” (line 343).

Discussion about the magma strain rate in reservoir is, to me, not important for the topic of such manuscript (as mentioned in the review letter).

Ok, we leave this part of the manuscripts as it is.

Reviewer #2

I had appreciated the first version of the manuscript, and raised a number of points that could improve it to the level required for a Nature Geoscience publication. I understand the authors have put substantial efforts in taking into account all of the comments and suggestions they received from all reviewers. That includes many new calculations referring to additional conditions, in order to demonstrate general applicability of their results. However, sincerely, I must conclude that the final result is much worse than the original. The ms has literally exploded, with 7 pages and 3 figures more (excluding methods), and a largely exceeding amount of details that are not suited for the main text - even an expert gets lost in too many numbers, conditions, exceptions, possibilities, etc.

We appreciated the comments of the reviewer during the first round of peer review, and we did our best to incorporate in the revised version of the manuscript all the suggestions he provided. We also tried to account for the comments of the reviewer #1 and #3 remaining in the editorial constraints requested by Nature Communications. We agree with the reviewer that the length of the manuscript and number of figures are exceeding the standards of Nature Geoscience, but these limitations are not relevant for our manuscript, being submitted for consideration to Nature Communications.

I still believe the ms contains interesting and relevant results that may deserve publication in NG. My suggestion is that the authors try to get back to the original format, adding the information required in the simplest possible form, and moving the details on the many conditions explored and the many calculations done to the methods. The main text should be accessible and interesting for a broad audience; in the present format, it is not.

After the first round of peer review, we did our best to make the manuscript accessible and appealing to a broad audience. Now, in the attempt to improve the average rate of the manuscript, as also suggested by the editor, we asked a native English colleague (a geologist/petrologist; Dr. Rebecca Astbury) to go through it checking the readability and the pleasing for a broad audience.

Reviewer #3

the authors have responded to my comments to my satisfaction

We thank the reviewer for the comment.